# TMEM16 scramblases thin the membrane to enable lipid scrambling

Maria E. Falzone[1,2,6], Zhang Feng [1,6], Omar E. Alvarenga[1,3], Yangang Pan[1], ByoungCheol Lee[1,5], Xiaolu Cheng [4], Eva Fortea[1,3], Simon Scheuring [1] & Alessio Accardi [1,2,4✉]

TMEM16 scramblases dissipate the plasma membrane lipid asymmetry to activate multiple eukaryotic cellular pathways. Scrambling was proposed to occur with lipid headgroups moving between leaflets through a membrane-spanning hydrophilic groove. Direct information on lipid-groove interactions is lacking. We report the 2.3 Å resolution cryogenic electron microscopy structure of the nanodisc-reconstituted $Ca^{2+}$-bound afTMEM16 scramblase showing how rearrangement of individual lipids at the open pathway results in pronounced membrane thinning. Only the groove's intracellular vestibule contacts lipids, and mutagenesis suggests scrambling does not require specific protein-lipid interactions with the extracellular vestibule. We find scrambling can occur outside a closed groove in thinner membranes and is inhibited in thicker membranes, despite an open pathway. Our results show afTMEM16 thins the membrane to enable scrambling and that an open hydrophilic pathway is not a structural requirement to allow rapid transbilayer movement of lipids. This mechanism could be extended to other scramblases lacking a hydrophilic groove.

[1] Department of Anesthesiology, Weill Cornell Medical College, New York, NY, USA. [2] Department of Biochemistry, Weill Cornell Medical College, New York, NY, USA. [3] Physiology, Biophysics and Systems Biology Graduate Program, Weill Cornell Medical College, New York, NY, USA. [4] Department of Physiology and Biophysics, Weill Cornell Medical College, New York, NY, USA. [5] Present address: Neurovascular Unit Research Group, Korea Brain Research Institute (KBRI), Daegu 41062, Republic of Korea. [6] these authors contributed equally: Maria E. Falzone, Zhang Feng. ✉email: ala2022@med.cornell.edu

**B**iological membranes play a fundamental role in many cellular signaling pathways as they define the physical boundaries of cellular compartments and actively modulate the function of integral and membrane-associated proteins. In eukaryotic cells, the composition and distribution of the phospholipid constituents of the membrane is tightly regulated by the activity of a variety of dedicated enzymes, flipases, flopases and scramblases[1]. The headgroup asymmetry of the plasma membrane is established by the action of ATP-driven pumps which distribute phosphatidylethanolamine (PE) and phosphatidylserine (PS) to the inner leaflet and phosphatidylcholine (PC) to the outer leaflet[1]. Activated phospholipid scramblases dissipate this asymmetry and expose PS on the extracellular leaflet. This is critical for multiple signaling pathways, ranging from apoptosis to blood coagulation, autophagy and cell-cell fusion[1,2]. There are five known families of scramblases, the $Ca^{2+}$-activated TMEM16[3–5], the caspase-activated Xk-related (Xkr) proteins[6], and the more recently identified ATG9, TMEM41B and VMP1[7–9]. Additionally, several GPCR's scramble lipids when reconstituted in liposomes[10,11].

Lipid scrambling by the TMEM16's is of critical importance for a myriad of physiological processes, including blood coagulation, bone mineralization, membrane fusion and viral entry[2,4,12]. Dysregulation of TMEM16 scramblase activity can have disastrous consequences, as both gain- and loss- of function mutations have been associated with disorders of blood, brain, bone and muscle[3,13–16]. The TMEM16 superfamily is comprised of Cl-channels and dual function scramblases/non-selective ion channels[4]. Both subtypes share a common homodimeric architecture where each protomer is comprised of 10 transmembrane (TM) helices[17–23] (Fig. 1A, B). In each protomer, the TM3-TM7 helices form a hydrophilic permeation pathway, or groove, that can adopt multiple conformations to allow passage of ions, lipids or to prevent movement of both substrates[20–24].

The scrambling mechanism has been extensively investigated at the functional, computational and structural levels[3,11,13,20–36]. The consensus proposal is a 'credit-card' mechanism[37], where the lipid headgroups penetrate and traverse the open hydrophilic groove while their tails remain embedded within the hydrocarbon core of the membrane[25,29,32]. Within this framework, lipid scrambling is enabled by specific interactions of the permeating lipids with charged and polar groove-lining residues[25,29,32]. However, TMEM16 scramblases do not discriminate among lipids such as PS, PE, PG, PC and DOTAP with headgroups differing in in charge, structure and size[18,26,28,30]. Further, PE lipids conjugated to cargoes of up to 5 kDa molecular weight are also efficiently scrambled[11]. These observations suggest that specific interactions between the groove and the scrambled lipids might not be necessary. The poor headgroup selectivity is also shared by other scramblases that lack an explicit hydrophilic groove, such as the GPCR opsin[10,11] and XKR8 and 9[6,38,39]. Moderate resolution structures[20,22] and molecular dynamics simulations[25,32] of fungal afTMEM16 and nhTMEM16 in nanodiscs showed these scramblases thin the membrane near the open groove, suggesting this might be important for lipid scrambling. Membrane thinning was also observed near the closed pathway of mTMEM16F, leading to the proposal that scrambling can also occur outside a closed groove[21]. Thus, it is not clear whether an open hydrophilic groove is required for scrambling. Direct structural information on how TMEM16 scramblases interact with lipids is essential to elucidate the molecular mechanisms of lipid permeation.

Here we use cryogenic electron microscopy (cryoEM) to determine the 2.3 Å resolution structure of the afTMEM16 scramblase from *Aspergillus fumigatus* in lipid nanodiscs. Our structure allows the direct visualization of lipids associated with the protein at the open groove and reveals that afTMEM16 thins the membrane at the open pathway by ~50%. The closest point of approach of the two membrane leaflets occurs near the wide intracellular vestibule of the groove, and no lipids could be resolved inside or interacting with the extracellular portion of the pathway. Mutagenesis of groove-lining residues does not perturb function, suggesting that specific interactions of permeating lipids with groove-lining residues are not essential for scrambling. We show that in thicker membranes scrambling is inhibited, while the groove remains in an open conformation. Conversely, in thinner membranes scrambling is enhanced although the groove is closed. Thus, lipid permeation is not always enabled by an open groove or prevented by a closed pathway. Based on these findings we propose that when the groove is open, the thinned membrane and the hydrophilic nature of the pathway synergistically lower the energy barrier for lipid scrambling. When the groove is closed, scrambling can occur, but at reduced rates in bilayers with plasma-membrane like thickness. In thinner membranes, closed-groove scrambling is enhanced allowing for lipid transport in the absence of $Ca^{2+}$.

## Results

**Structural basis of lipid reorganization by the afTMEM16 scramblase.** To gain insight into how the afTMEM16 scramblase alters the organization of the membrane and interacts with the surrounding lipids we used cryoEM to determine its structure in the $Ca^{2+}$-bound conformation in nanodiscs at 2.3 Å (Fig. 1, Supplementary Fig. 1). Nanodiscs were formed using the MSP1E3 scaffold protein and a 7:3 mixture of 1,2-Dioleoyl-sn-glycero-3-phosphocholine (DOPC, 18:1 PC) and 1,2-Dioleoyl-sn-Glycero-3-Phosphatidylglycerol (DOPG, 18:1 PG), termed C18 lipids (Table 1). In saturating 0.5 mM $Ca^{2+}$, conditions referred to as C18/$Ca^{2+}$, afTMEM16 is maximally active[20], therefore we hypothesize this represents the active state of the scramblase. The present structure is nearly superimposable to the previously determined $Ca^{2+}$-bound structure of afTMEM16 in a 3:1 mixture of 1-palmitoyl-2-oleoyl-sn-glycero-3-phosphoethanolamine (POPE, 16:0–18:1 PE) and 1-palmitoyl-2-oleoyl-sn-glycero-3-phospho-(1′-rac-glycerol) (POPG, 16:0–18:1 PG) nanodiscs[20], with Cα rmsd ~0.8 Å. Thus, headgroup choice and acyl-chain saturation do not influence the conformation of the protein. The significantly improved resolution of the C18/$Ca^{2+}$ map allowed us to resolve 4 water molecules in the $Ca^{2+}$ binding sites which coordinate two bound ions (Supplementary Fig. 2B), bringing the respective coordination number of the bound $Ca^{2+}$ ions to 7 and 8, consistent with the high affinity of these sites[30]. The map also contains non-protein densities that could be modeled as lipids associated with the protein (Fig. 1C–J, Supplementary Fig. 2). To improve the quality of the density of the lipids near the pathway, we carried out symmetry expansion and additional rounds of 3D classification, which yielded one class with an additional four resolved lipids (Supplementary Fig. 1E), for a total of 32 resolved lipids, 16 in each monomer (Fig. 1H–J, Supplementary Fig. 2). The observed lipids define the nearly continuous interfaces of the scramblase with the inner and outer membrane leaflets near the dimer interface (lipids D1-D9) and illustrate how the poses adopted by individual lipids result in the profound remodeling of the membrane induced by afTMEM16 near the lipid pathway (lipids P1-P7) (Fig. 1H–J).

**Lipids form a cap around the transmembrane dimer interface.** The transmembrane dimer interface of afTMEM16 is formed by the extracellular half of TM10 from each monomer (Fig. 1K–L, Supplementary Fig. 2D, E). This minimal interface contains several hydrophobic residues and two membrane-embedded salt bridges formed by E618 and H619 of opposite subunits positioned ~1/3 of the way through the membrane from the extracellular leaflet (Fig. 1K–L). In the C18/$Ca^{2+}$ structure, these salt

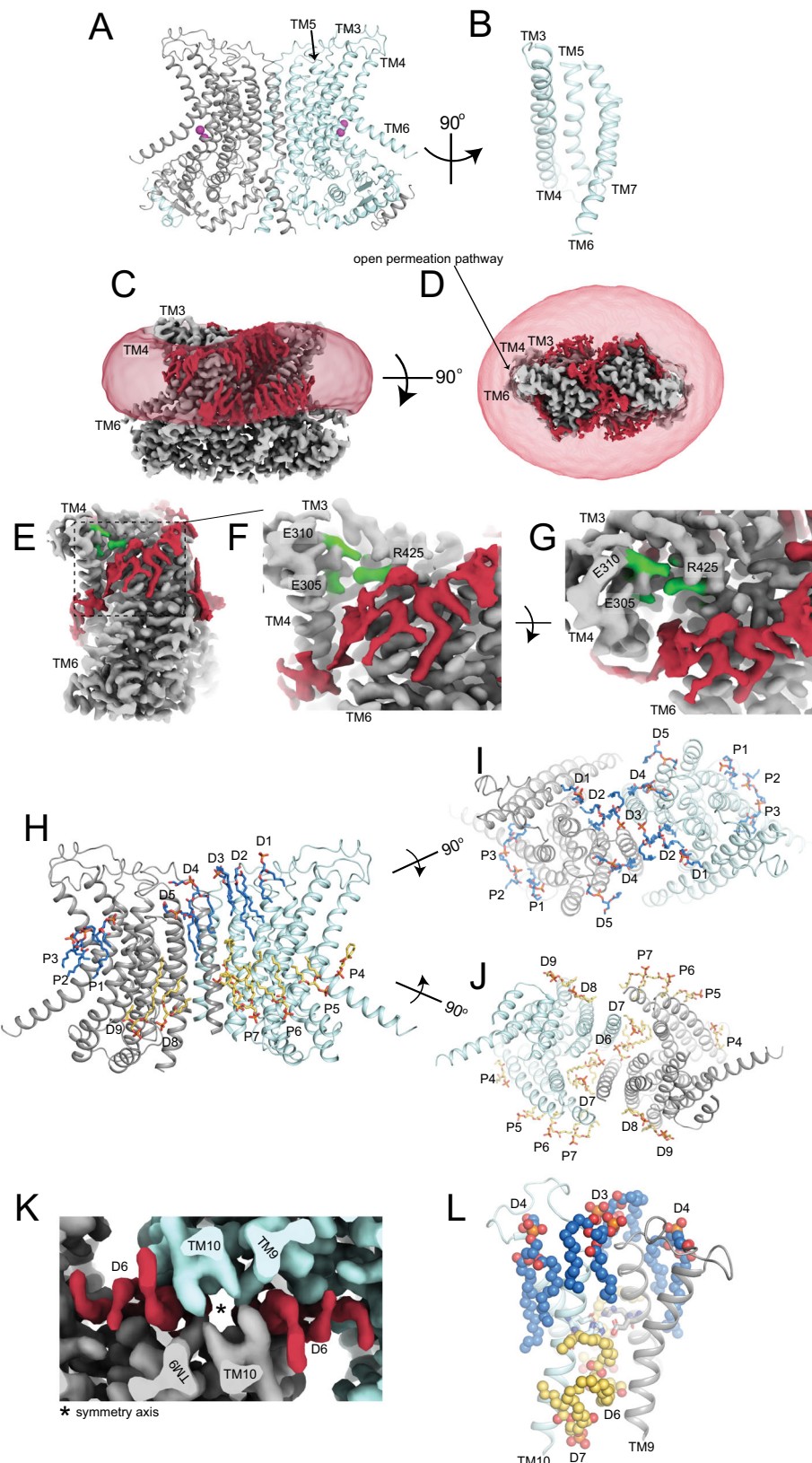

bridges appear to be isolated from the intra- and extra-cellular solutions by eight well-defined lipids (D3, D4, D6 and D7 from each subunit), four above and four below the interacting residues (Fig. 1L). On the extracellular side, the D3 lipids from opposite subunits straddle the N terminal region of TM10 with their heads positioned by the side chains of C607 and W608 to make direct

contact above the symmetry axis (Supplementary Fig. 2D). A second lipid, D4, is wedged between TM9 and TM10 with its head coordinated by polar and charged residues in the TM9–10 linker (N593, P598, T604 and R606; Supplementary Fig. 2D). On the intracellular side, the heads of D6 from each subunit make contact across the symmetry axis and are wedged between the

**Fig. 1 Lipid-protein interactions in Ca²⁺-bound afTMEM16. A** Structural model of afTMEM16 in 0.5 mM Ca²⁺ in C18 lipid nanodiscs. **B** View of the open permeation pathway. **C**, **D** Unsharpened maps of the protein (gray) and associated lipids (red) viewed from the membrane plane (**C**) and from the extracellular side (**D**). The map showing the density of the nanodisc membrane is low-pass filtered to 10 Å and shown in transparent red. **E**–**G** Views of the lipid groove from the plane of the membrane (**E**, **F**) and from the extracellular solution (**G**). The unsharpened maps of the protein (gray) and lipids (red) are shown. **H**–**J** Views of the afTMEM16 dimer from the plane of the membrane (**H**), extra- (**I**) and intra-cellular (**J**) sides with modeled lipids shown in stick representation. Lipids at the dimer interface are labeled D1–9 and those at the permeation pathway are labeled P1-7. Lipids from the inner and outer leaflets are colored in yellow and blue, respectively. Lipids D4 and D5 were built as PG, while all others were built up to the phosphate atom in the head. In all views, the cytosolic domain of afTMEM16 was omitted for clarity. **K** Close up of the density map at the dimer interface viewed from the extracellular side showing the two afTMEM16 monomers (gray and cyan) and intercalated lipid tails (red). * denotes the symmetry axis. **L** The dimer interface salt bridge between TM9 and 10 (in cartoon representation) is formed by E618 and H619 (in stick representation) and is shielded from the intra- and extra-cellular solutions by lipids D3, D4, D6, and D7 (in spheres and colored as in **F**–**H**).

C-termini of the TM10s (Supplementary Fig. 2E). They are coordinated by D571, G574 and W578 on the TM9 from one subunit and by R625, Y626 and R629 from TM10 on the other (Supplementary Fig. 2E). Additionally, the head of D7 is coordinated by Y626, S630 and K634 from TM10 of one subunit and by Q364 on TM5 and D571 on TM9 from the opposite subunit (Supp. Figure 2E). The tails of these 8 lipids are accommodated in hydrophobic grooves between TM2, 9, 10 from both subunits (Fig. 1K, Supplementary Fig. 2D, E). The intercalated organization of the lipid tails and helices gives rise to densely packed hydrophobic regions that shield the interacting E618 and H619 residues from water access, possibly strengthening their electrostatic interaction (Fig. 1L). These observations, together with the evolutionary conservation of the E618/H619 pair (Supplementary Fig. 2C) and of the TMEM16 fold suggests these lipids might play a structural role in stabilizing the dimeric architecture of all TMEM16s. Finally, lipids in the dimer cavity are oriented nearly perpendicularly to the plane of the membrane and the heads of D3 straddle the TM10 dimer interface (Fig. 1K–L). This arrangement results in an local thickening of the membrane at this hydrophobic region, supporting the idea that scrambling does not occur at the dimer cavity[18].

**Structural basis of membrane thinning at the scrambling pathway.** The C18/Ca²⁺ structure reveals how the scramblase reorients the lipids that approach the open scrambling pathway (Figs. 1E–G and 2A). Near the dimer interface, the planes of the outer (OL) and inner (IL) leaflets are respectively defined by lipids D1–4 and D8–9, in good agreement with the outline of the low pass filtered nanodisc map (Fig. 1C). The downward slope of the OL starts at D5, a well-defined PG lipid (Figs. 1H, 2B, Supplementary Fig. 2A), and progresses towards the open groove as P1 and P2 adopt distorted poses with their headgroups becoming increasingly tilted (Fig. 2A, B). The IL bends upwards and P5-P7 become increasingly tilted as their heads climb around the intracellular portions of TM3 and TM4, coordinated by the hydrophilic side chains of T341, K345 and T334 (Fig. 2A, C). At the pathway, P3 is sandwiched between TM4 and TM6 near the constriction formed by T325 and Y432 and its headgroup points towards the extracellular side, such that it is contiguous to other OL lipids (Fig. 2A). Notably, P3 and P4 are oriented with their heads facing the outside of the groove (Fig. 2A). The distance between the phosphate atoms of the heads of P3 and P4 in the OL and IL is <22 Å (Fig. 2A), showing that the hydrocarbon core of the membrane is thinned by ~50% at the open pathway. A similar thinning is seen in the low-pass filtered nanodiscs map near the pathway (Fig. 1C, D).

**Lipids outside the open pathway define the distorted membrane interface.** The identification of sites where lipids bind at or near the open groove raises the possibility that scrambling could occur via a 'conveyor belt' mechanism, where lipids translocate between leaflets by moving from site to site. Alternatively, the observed lipids could define the protein-membrane boundary but not necessarily be translocated, with the possible exception of P3 within the pathway (Fig. 2A). To distinguish between these hypotheses, we investigated how mutating residues coordinating the resolved portions of the headgroups of P1–2 and P4–6 impacts scrambling. We found that mutations aimed at disrupting the headgroup interactions of P1-P2 (W202A/R427A/I431A/W529A), P4-P5-P6 (R279A/T334A/K345A/Y349A) or P2-P5-P6 (R279A/K345A/R427A/K428A) have minimal functional effects in both saturating and 0 Ca²⁺, less than 5-fold reduction in scrambling (Fig. 2D, E, Supplementary Fig. 3). This suggests that these lipid sites are not obligatory on the path taken by scrambled lipids. Rather, other factors, such as tail interactions with interhelical grooves, might contribute to their association with afTMEM16 (Supplementary Fig. 2F–G) and stabilize the distorted membrane-protein interface that results in thinning at the pathway.

**Scrambling does not require specific interactions with extracellular groove-lining residues.** One unexpected feature of our structure is that the extracellular vestibule of the groove does not directly interact with the membrane and no lipids could be resolved there (Fig. 1E–G), as they cross the open groove at the T325 and Y432 constriction (Fig. 2A). The head of the P3 lipid faces the outside of the groove and its phosphate is too far to interact with the charged side chains of E305 (13.8 Å), E310 (17.9 Å) and R425 (15.7 Å) that line the extracellular groove (Figs. 1E–G and 3A). We mutated 16 side chains that line the central constriction or the extracellular vestibule of the groove: I298, F302, E305 and E310 on TM3; K317, Y319, F322, T325 and I332 on TM4; T373 and S374 on TM5; and R425, K428, Q429, Y432 and F433 on TM6 (Fig. 3A–C). In saturating Ca²⁺, single or multiple simultaneous alanine substitutions at these sites have only minor effects on scrambling, less than 2-fold reduction in rates (Fig. 3D). In 0 Ca²⁺, some mutations have slightly larger effects, but less than 7-fold in all cases (Fig. 3E, Supplementary Fig. 3). Thus, scrambling does not require specific interactions of lipids with residues lining the extracellular vestibule or the central constriction of the groove.

The wide intracellular vestibule is embedded in the nanodisc membrane and, at the open pathway, the P3 and P4 lipids have opposite orientations (Fig. 2A), suggesting scrambling might occur in the groove region comprised between them. In this case, the lipid headgroups would only need to move through the wide intracellular vestibule of the pathway below the T325-Y432 constriction (Fig. 2A). We propose that the observed membrane thinning acts in concert with the hydrophilic environment of the open groove to enable scrambling by lowering the energy barrier for lipid movement. This hypothesis is supported by the modulation of scrambling activity by afTMEM16 and hTMEM16K by lipid acyl chain length[20,28]. These effects should

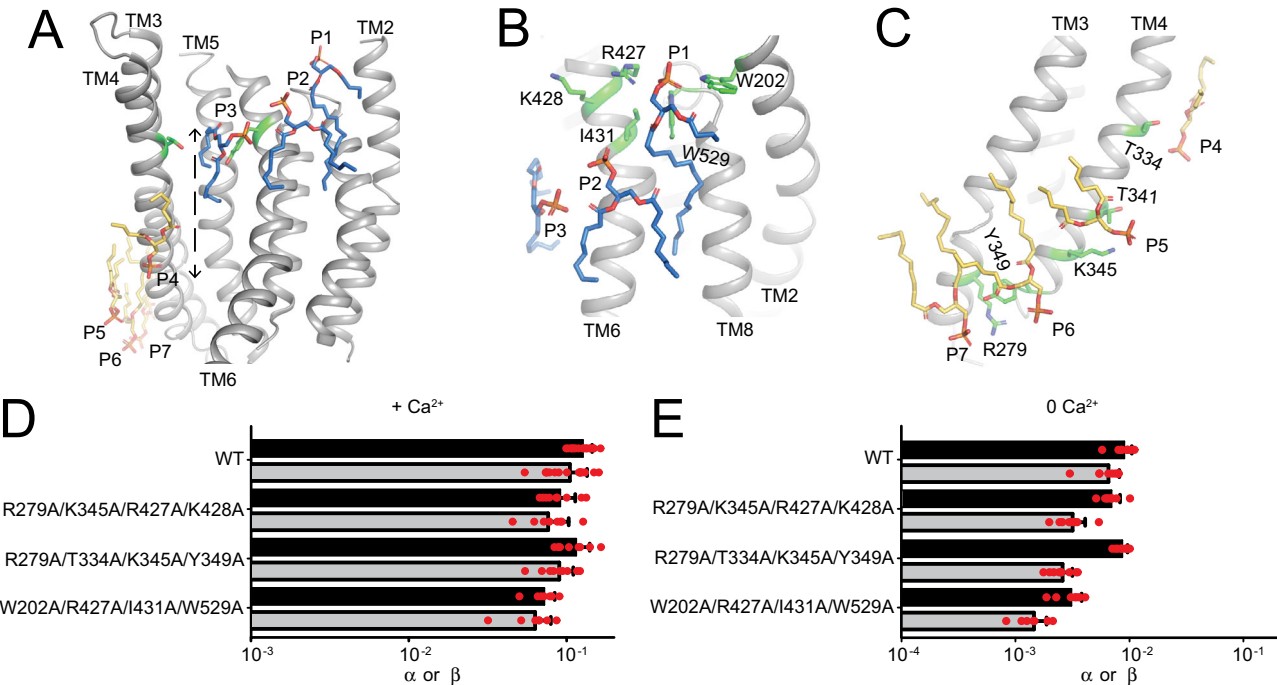

**Fig. 2 Coordination of lipids outside the permeation pathway. A** View of the seven pathway lipids (in sticks, colored as in Fig. 1H). T325 and Y423 are shown as green sticks. Dashed arrow indicates the distance between the phosphate atoms of the last lipid from the inner (P4) and outer (P3) leaflets. **B, C** Coordination of P1-P2 (**B**) and of P4-P7 (**C**). Side chains are shown in green sticks. **D, E** forward ($\alpha$) and reverse ($\beta$) scrambling rate constants for indicated quadruple mutants of residues coordinating lipids outside the pathway (P1-2 and P4-7). Bars are average values for $\alpha$ (black) and $\beta$ (gray) determined from $n = 7$–16 independent determinations from $N = 2$–8 independent preparations, error bars are S. Dev., and red circles are values from individual repeats. Source data is available in the Source Data file.

**Table 1 AFM and cryoEM determination of membrane thickness for considered lipid compositions.**

| Name | Lipid component chain length (70% PC:30%PG) | Height from AFM (nm) | Height estimated from EM maps (nm) |
|---|---|---|---|
| C14 | 50% 14:0, 50% 16:0–18:1 | 3.2 ± 0.3 | 3.0 ± 0.2 |
| C16 | 100% 16:1 | 3.2 ± 0.2 | N.d. |
| C18 | 100% 18:1 | 3.4 ± 0.2 | (0 Ca$^{2+}$) 3.1 ± 0.4 |
| C18 | 100% 18:1 | 3.4 ± 0.2 | (0.5 mM Ca$^{2+}$) 3.3 ± 0.1 |
| C20 | 100% 20:1 | 3.7 ± 0.2 | N.d. |
| 70% C22 | 70% 22:1, 30% 18:1 | 4.0 ± 0.3 | N.d. |
| 70% C22 | 70% 22:1 PC, 30% 18:1 PG | 4.0 ± 0.2 | N.d. |
| C22 | 100% 22:1 | 4.1 ± 0.2 | 3.9 ± 0.1 |

Heights were estimated using AFM tomography, reported values indicate the peak FWHH ± of the value distribution (see Supplementary Fig. 4A, B and Methods). Membrane height from cryoEM was determined from C1 unsharpened EM maps using the difference in z coordinate for the inner and outer leaflet at (x,y) far from the protein. Reported values are the average ± S.Dev of 3 different points. N.d. indicates compositions for which no cryoEM data was determined.

reflect the ability of the scramblase to sufficiently thin these membranes, rather than arise from lipid-dependent conformational changes of the groove.

**Regulation of lipid scrambling by membrane thickness**. We measured how systematic variation of lipid acyl chain length affects lipid scrambling by afTMEM16. We kept the lipid headgroup composition at a constant ratio of 7 PC: 3 PG and used acyl chains with a single unsaturation and 16–22 carbons, C16-C22 lipids (Table 1). Liposomes formed from a 7:3 mix of 14:1 PC and PG lipids were not stable in our scrambling assay (Supplementary Fig. 4C). We generated thinner membranes using a 50-50 mixture comprised of short chain1,2-dimyristoyl-sn-glycero-3-phosphocholine (DMPC, C14:0) and 1,2-dimyristoyl-sn-glycero-3-phospho-(1'-rac-glycerol) (DMPG, C14:0) and of POPC and POPG[28]. A 7:3 PC:PG headgroup ratio was

maintained also in this mix, referred to as C14 (Table 1). Atomic force microscopy (AFM) measurements show that membrane thickness varies between ~3.2 nm and ~4.1 nm, with near-linear scaling with acyl chain length (Table 1, Supplementary Fig. 4A, B).

In the presence of saturating 0.5 mM Ca$^{2+}$ the scrambling rate constants do not depend on membrane thickness between ~3.2 nm (C14 lipids) and ~3.9 nm (C20 lipids) (Fig. 4A). In contrast, scrambling is nearly completely inhibited in C22 lipids Fig. 4A)[20]. Thus, in saturating Ca$^{2+}$ there is chain length selectivity with a threshold for activity below membrane thickness of ~4.1 nm. In contrast, in 0 Ca$^{2+}$ scrambling displays a nearly exponential inverse dependence on membrane thickness (Fig. 4B). Remarkably, in C14 lipid membranes scrambling by afTMEM16 is nearly Ca$^{2+}$-independent, with rate constants only ~3-fold lower in 0 Ca$^{2+}$ compared to the ~20-fold reduction seen in C18 membranes (Fig. 4A, B, Supplementary Fig. 4).

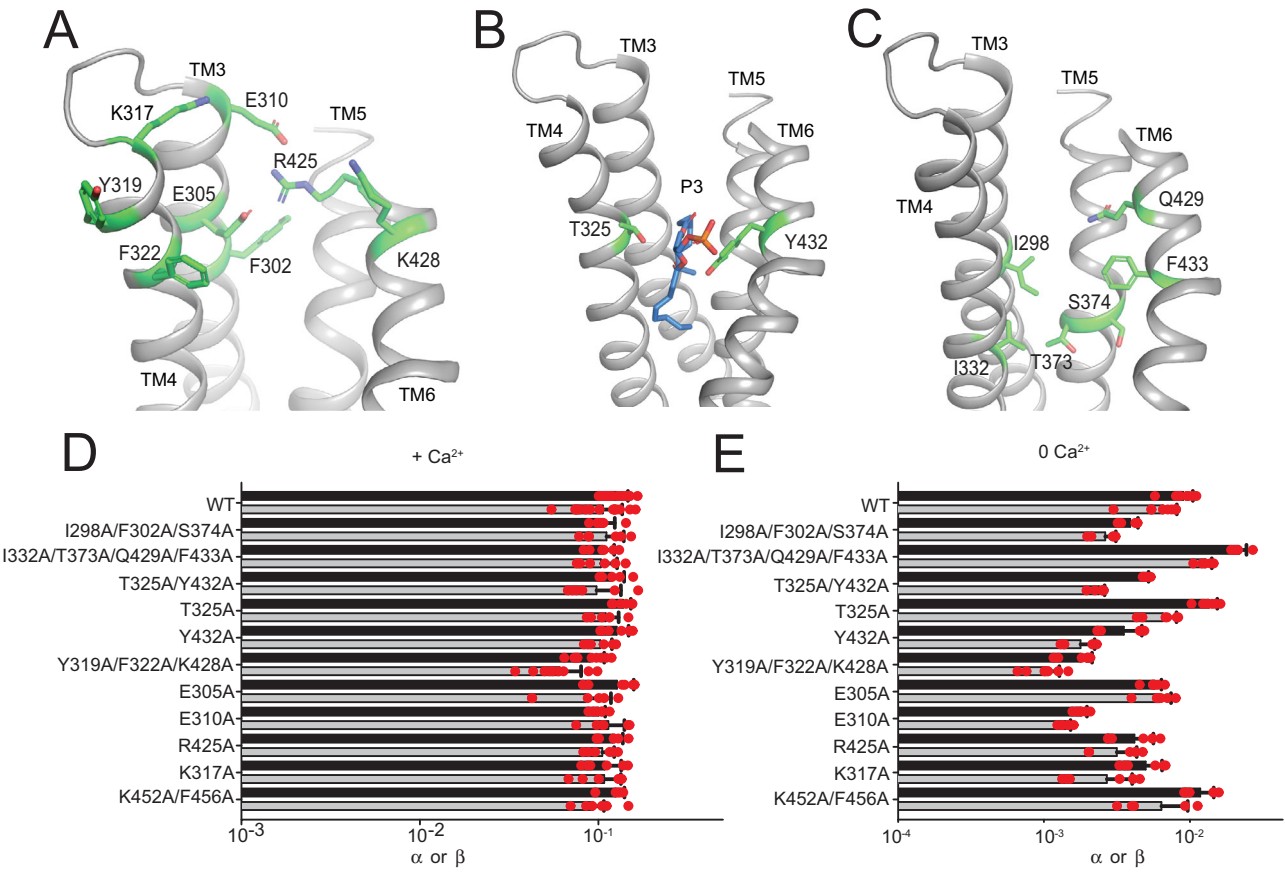

**Fig. 3 Functional role of groove-lining residues in lipid scrambling. A–C** Residues lining the extracellular vestibule (**A**), coordinating P3 (**B**) and lining the central constriction (**C**) are shown as green sticks. **D**, **E** Forward (α) and reverse (β) scrambling rate constants of single and multiple alanine substitutions at the indicated positions. Bars are average values for α (black) and β (gray) determined from $n = 5–16$ independent determinations from $N = 2–8$ independent preparations, error bars are S. Dev., and red circles are values from individual repeats. Source data is available in the Source Data file.

To test whether the long C22 acyl chains inhibit scrambling in saturating Ca²⁺ by occluding the pathway[40] we measured scrambling in membranes formed by 70% C22 lipids and 30% C18 lipids, which are ~4 nm thick (Table 1). In saturating Ca²⁺ scrambling activity is similar to that seen in 100% C18 lipids (Fig. 4A), while in 0 Ca²⁺ there is a ~17-fold reduction, consistent with the reduction expected for membranes of this thickness (Fig. 4B). This behavior does not depend on whether the mixed chain lengths were segregated by headgroup (Fig. 4A, B). Thus, the tails of C22 lipids are not 'blockers' of the afTMEM16 permeation pathway. These results suggest that in 0 Ca²⁺ scrambling rates are proportional to the energetic cost of lipid headgroups crossing the hydrophobic core of the membrane, while in the presence of Ca²⁺ other factors contribute to scrambling.

**Ca²⁺-bound afTMEM16 has an open groove in C22 membranes.** To determine whether the C22 lipids inhibit scrambling by favoring a closed groove conformation (Fig. 4C), we determined the 2.7 Å cryoEM structure of nanodisc-reconstituted afTMEM16 in the presence of saturating Ca²⁺ (Fig. 4D, Supplementary Fig. 5A–G). Despite a ~500-fold reduction in scrambling activity the groove remains open in a conformation nearly identical to that seen in C18 lipids, Cα r.m.s.d. ~0.35 Å (Fig. 4D). Neither the C18/Ca²⁺ nor the C22/Ca²⁺ datasets display structural heterogeneity as no additional classes could be identified using multiple rounds of iterative 3D classifications on afTMEM16 dimers and monomers using different classification parameters and softwares (see Methods, Supplementary Figs. 1, 5,

7, 8, 10). The 3.5 Å resolution structure of afTMEM16 in the larger MSP2N2 nanodiscs in C22/Ca²⁺ (Supplementary Fig. 5H–N) shows an open permeation pathway in all 3D reconstructions, with Cα r.m.s.d. ~0.5 Å to C18/Ca²⁺ and ~0.4 Å to C22/Ca²⁺_{MSP1E3} (Fig. 4D). Thus, in afTMEM16 an open groove is not sufficient to enable lipid scrambling and nanodisc size does not influence the conformation.

In the C22/Ca²⁺ maps we resolved several lipids near the dimer interface corresponding to D2, D3, D6, and D7 (in MSP1E3 map) and to D2 and D6 (in MSP2N2 map) seen in the C18/Ca²⁺ map (Supplementary Fig. 6). In the MSP1E3 map we detected strong density for P6 and P7, located near the intracellular loop connecting TM3 and TM4 (Supplementary Fig. 6). However, despite the high resolution of the C22/Ca²⁺ MSP1E3 map, we detect only weak signals for lipids associated with the pathway-delimiting helices TM4 and TM6. This suggests that the interactions of C22 lipids with the pathway helices are weaker than those of C18 lipids, possibly reflecting a higher energy cost associated with distorting these longer acyl chain lipids.

**Scrambling in 0 Ca²⁺ does not require groove opening.** The finding that an open groove is not sufficient to allow lipid movement raises the question of whether a closed groove prevents lipid scrambling entirely. Many proteins that scramble lipids lack explicit membrane-exposed hydrophilic grooves[10,38,39,41] and most purified TMEM16's scramble lipids in 0 Ca²⁺ when the groove is predominantly closed[18,28,30]. This basal activity could reflect transient openings of the pathway, however an open groove

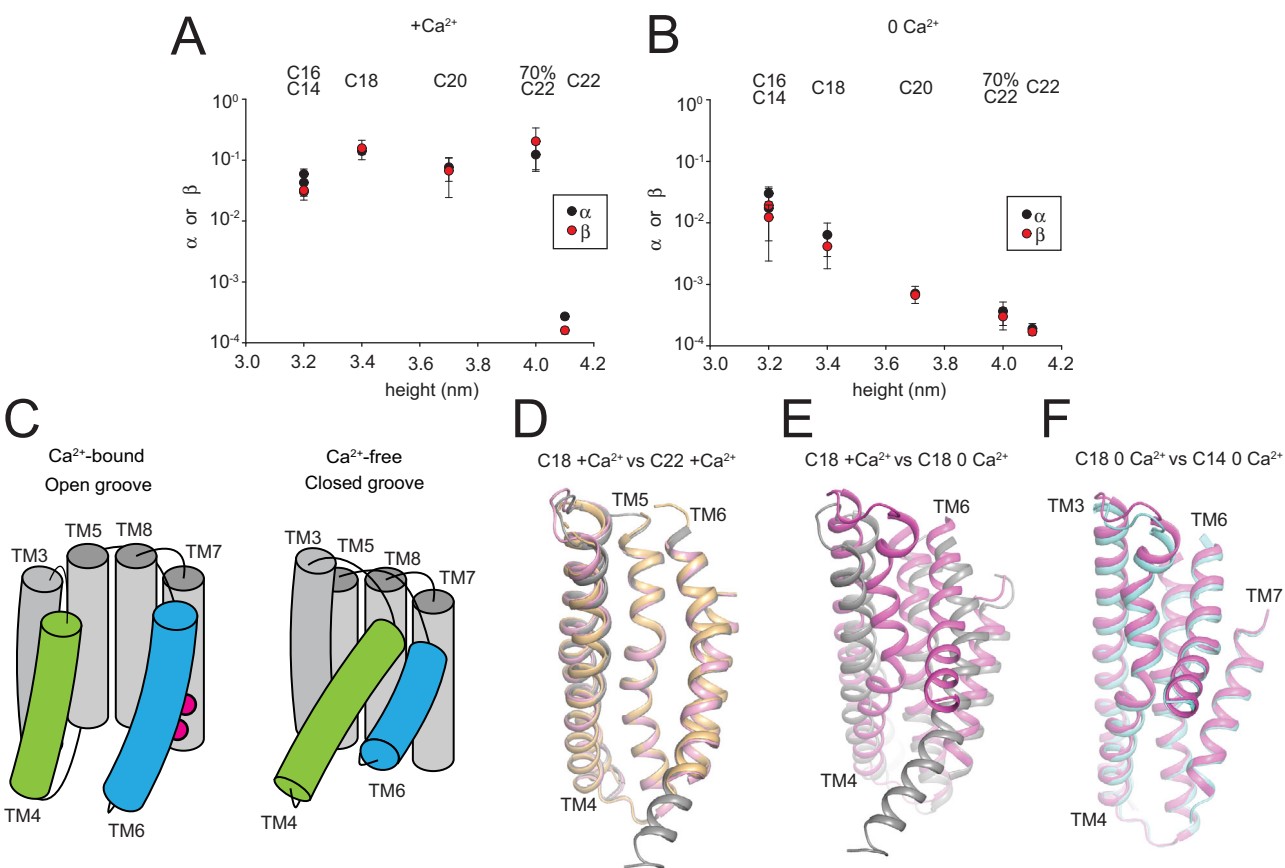

**Fig. 4 Functional and structural regulation of lipid scrambling by membrane thickness. A, B** Forward ($\alpha$, black circles) and reverse ($\beta$, red circles) scrambling rate constants as a function of membrane thickness in the presence of 0.5 mM (**A**) or 0 $Ca^{2+}$ (**B**). Values are the mean determined from $n = 8$–15 independent determinations from $N = 3$–5 independent preparations and error bars represent St. Dev. Corresponding lipid compositions are noted above. Values for C18 and C22 were determined in another study and are reported here for comparison[20]. Source data is available in the Source Data file. **C** Cartoon representation of the permeation pathway showing movements of TM4 (green) and TM6 (blue) during $Ca^{2+}$-gating. **D–F** Alignment of the permeation pathway of afTMEM16 in (**D**) in 0.5 mM $Ca^{2+}$ and C18 (gray), C22 MSP1E3 nanodiscs (light pink) and C22 MSP2N2 nanodiscs (orange), (**E**) C18 nanodiscs in 0.5 mM (gray) and 0 $Ca^{2+}$ (pink), (**F**) in 0 $Ca^{2+}$ in C18 (red) and C14 nanodiscs (cyan). Source data are provided as a Source Data file.

$Ca^{2+}$-free conformation has not been observed in a membrane environment[20–23]. Alternatively, TMEM16 scramblases could thin the membrane enough to enable slow lipid scrambling outside of a closed groove, as proposed for the mammalian TMEM16F[21].

To elucidate the structural bases of scrambling in the absence of $Ca^{2+}$, we determined the 3.1 Å resolution structure of afTMEM16 in C18 lipids in 0 $Ca^{2+}$ (Fig. 4E, Supplementary Fig. 7). Extensive classification of afTMEM16 dimers and of symmetry-expanded monomers (see Methods) revealed only reconstructions corresponding to a closed groove conformation (Fig. 4E, Supplementary Fig. 7). In C18 lipids the basal scrambling activity of afTMEM16 is modest, ~4.5% of that in saturating $Ca^{2+}$ (Fig. 4A, B, Supplementary Fig. 4), suggesting that the fraction of particles that could adopt a $Ca^{2+}$-free open groove conformation could be too small to be detected. In contrast, in C14 lipids scrambling in 0 $Ca^{2+}$ is ~30% of that seen in saturating $Ca^{2+}$ (Fig. 4A, B, Supplementary Fig. 4) so that if scrambling requires groove opening, we expect that a significant portion of particles would adopt a $Ca^{2+}$-free open-groove conformation. Analysis of a C14/0 $Ca^{2+}$ afTMEM16 dataset yields only classes with a closed groove (Fig. 4F), Cα r.m.s.d. ~0.9 Å to C18/0 $Ca^{2+}$, the highest of which reached 3.3 Å average resolution (Supplementary Fig. 8). Thus, in 0 $Ca^{2+}$ changing from C18 to C14 lipids induces a ~8-fold increase in scrambling activity that is not accompanied by a concomitant increase in the

probability of opening of the groove. This suggests that the basal, $Ca^{2+}$ independent activity is due to closed-groove scrambling.

This hypothesis is further supported by the analysis of the D511A/E514A mutant of afTMEM16 that disrupts the $Ca^{2+}$-binding site. This mutation impairs TMEM16 activity by preventing opening of the pathway[13,30,42,43] and scrambles lipids in a $Ca^{2+}$-independent manner at rates comparable to those of the WT protein in 0 $Ca^{2+}$[30]. Scrambling by D511A/E514A afTMEM16 is modulated by membrane thickness like the WT protein in 0 $Ca^{2+}$ (Fig. 5A, Supplementary Fig. 9), and in C14 membranes its activity is ~30% of that of the WT protein in C18 lipids and saturating $Ca^{2+}$. To test whether the D511A/E514A afTMEM16 adopts an open-pathway conformation in conditions of high activity, we determined its structure in C14 nanodiscs with 0.5 mM $Ca^{2+}$ to 3.1 Å resolution (Fig. 5B, Supplementary Fig. 10). As expected, despite the presence of 0.5 mM $Ca^{2+}$, the protein adopts the same conformation as in the WT apo structure and there is no density in the $Ca^{2+}$ binding site. In all reconstructions the permeation pathway is closed, with Cα r.m.s.d. ~1.1 Å to C18/0 $Ca^{2+}$ and ~0.4 Å to C14/0 $Ca^{2+}$ (Fig. 5B, Supplementary Fig. 10), indicating that increased scrambling is again not accompanied by higher open probability of the groove. Together, our results suggest that scrambling of afTMEM16 in 0 $Ca^{2+}$ occurs outside of a closed groove. Calcium-independent openings of the lipid permeation pathway, if they occur, are

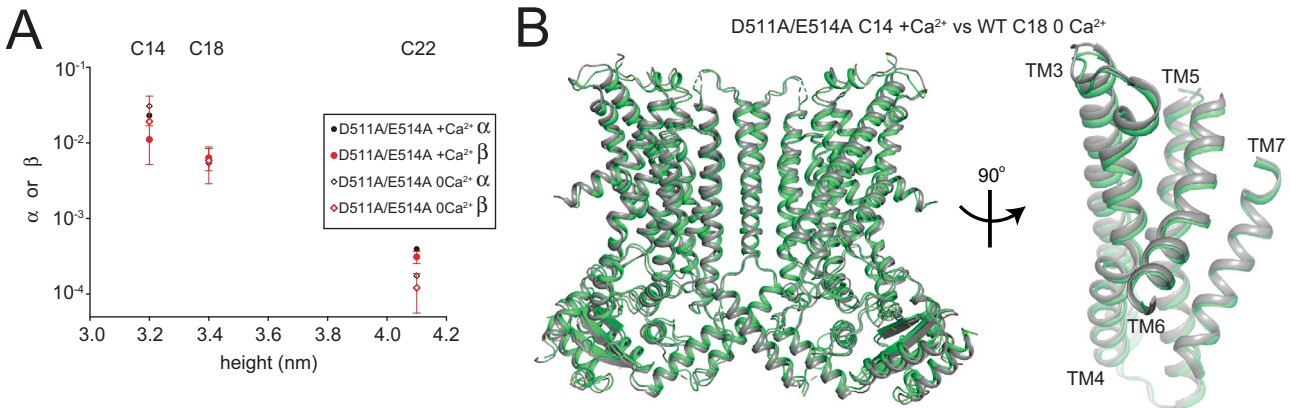

**Fig. 5 Functional and structural characterization of afTMEM16 D511A/E514A. A** Forward (α, black circles) and reverse (β, red circles) scrambling rate constants of D511A/E514A afTMEM16 in 0.5 mM (filled symbols) or 0 Ca²⁺ (empty symbols). Values are the mean determined from *n* = 7–9 independent determinations from *N* = 2–3 independent preparations and error bars represent St. Dev. Corresponding lipid compositions are noted above. Raw data is available in the Source Data file. **B** Alignment of afTMME16 D511A/E514A in the presence of Ca²⁺ (green) in C14 lipids with wildtype afTMEM16 in 0 Ca²⁺ in C18 lipids (gray) with close up view of the permeation pathway. Source data are provided as a Source Data file.

transient and cannot account for the observed increase in activity. Thus, an open groove is not necessary for lipid scrambling.

In the three datasets for apo afTMEM16 (C18/0 Ca²⁺, C14/0 Ca²⁺ and DA/EA in 0.5 mM Ca²⁺) we could resolve 4–9 lipids per monomer, all localized near the dimer interface in positions closely resembling those seen in C18/Ca²⁺ structure (Supplementary Fig. 11), supporting the notion that these lipids interact strongly with the protein. No lipids could be resolved near the closed pathway in these structures. The average resolution of these datasets is lower than that of the two Ca²⁺-bound structures, preventing us from drawing mechanistic inferences from this observation.

**Scrambling activity correlates with membrane thinning at the pathway.** Our proposal that afTMEM16 enables scrambling by thinning the membrane at the permeation pathway predicts there should be a correlation between thinning and function. Although a quantitative evaluation of thinning is precluded by the different resolutions of our maps, a qualitative analysis of the nanodisc density maps supports this notion (Fig. 6A–D, Supplementary Fig. 12). Far from the protein, membrane thickness of C14, C18 and C22 nanodiscs is comparable to that determined by AFM (Table 1). In the C18/Ca²⁺ map the membrane near the open groove appears markedly thinned (Fig. 6B) and closely tracks the position of individual lipids (Fig. 2A). Thinning is reduced near the open pathway of the C22/Ca²⁺ map (Fig. 6A, Supplementary Fig. 12 and near the closed pathway of the C18/0 Ca²⁺ map (Fig. 6C), consistent with the reduced scrambling activity (Fig. 4A, B). In the C14/0 Ca²⁺ map, thickness at the closed pathway qualitatively approaches that at the open pathway of the C18/Ca²⁺ map (Fig. 6D, Supplementary Fig. 12), consistent with enhanced scrambling (Fig. 4B). These qualitative observations suggest there is a direct correlation between the thickness of the membrane near the pathway and scrambling activity. This supports the idea that in C22 membranes scrambling could be inhibited because of the reduced thinning despite an open groove, and that the closed groove conformation of afTMEM16 is scrambling competent because it thins the membrane enough to enable lipid flipping.

## Discussion

Activation of scramblases catalyzes the rapid movement of phospholipids between membrane leaflets and results in the externalization of charged and polar lipids that trigger a variety of fundamental physiological processes[1,2,4]. The current consensus

is that TMEM16 scramblases mediate lipid transport via a credit-card like mechanism[37], with the headgroups forming specific interactions with polar and charged residues lining the full length of the hydrophilic groove[25,28,29,32]. This predicts that scramblases should discriminate among lipids based on their headgroups but not their tails, and that mutations of groove-lining residues should affect lipid scrambling. However, the TMEM16's[11,18,23,26,30], the Xkr's[6,44] and GPCR's moonlighting as scramblases[10] do not select among lipids with different headgroups non-selective scramblases. Further, both the Xkr's and GPCR's lack explicit hydrophilic grooves[10,38,39], bringing the structural requisites of the credit card mechanism into question.

Here, we combined structural and functional experiments to investigate the mechanism of lipid scrambling by the TMEM16's. The 2.3 Å structure of afTMEM16 reconstituted in C18 nanodiscs shows how individual lipids interact with the scramblase to define the thinned and distorted profile of the membrane near the open pathway (Fig. 1C–H). Resolved lipids mainly localize to the periphery of the groove, interacting with the intracellular portions of TM3-4 and with the extracellular portions of TM6-7. The position of the last lipids from the intra- and extra-cellular leaflets suggests that headgroup flipping occurs in the intracellular vestibule. No density for lipids was visible near the extracellular vestibule (Figs. 1 and 2A) and mutations of residues lining this narrow constriction, or the groove interior have no functional effects (Figs. 2 and 3). Reconstituting afTMEM16 in membranes formed from lipids with longer acyl chains dramatically inhibits scrambling although the groove remains open (Fig. 4). Conversely, reconstitution in thinner membranes facilitates scrambling even when the groove is closed (Figs. 4 and 5).

Together, these results have three important implications; first, lipid scrambling does not require specific interactions with the groove's hydrophilic interior or its extracellular vestibule. Second, acyl chains rather than headgroups are key determinants of scrambling activity (Fig. 4A, B). Third, an open groove is neither sufficient nor necessary for scrambling (Figs. 4 and 5). These findings are inconsistent with a credit-card mechanism. We propose that lipid scrambling is primarily determined by the ability of afTMEM16 to thin the membrane near the pathway, so that lipids mainly interact with the surface of the groove without penetrating deep within its narrow and hydrophilic interior (Fig. 7). The membrane-thinning mechanism readily explains evolutionarily conserved properties of TMEM16 scramblases that are difficult to reconcile with the credit-card mechanism, such as

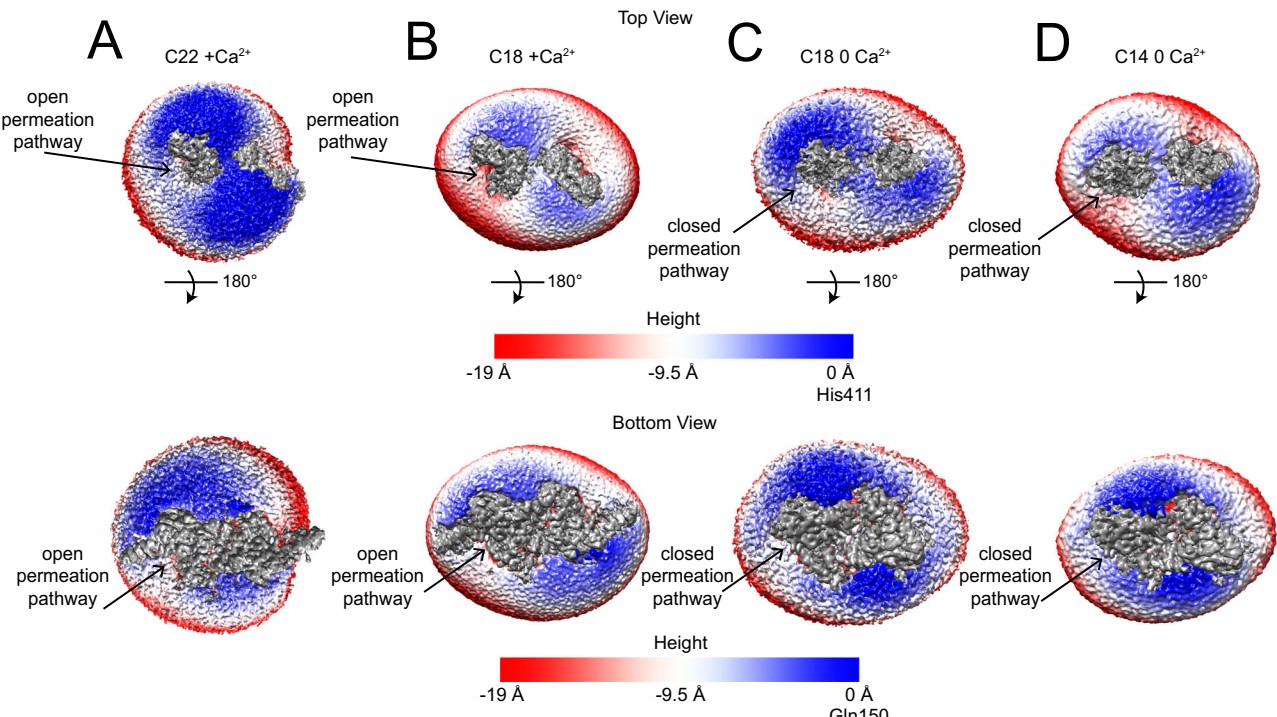

**Fig. 6 Membrane thinning at the afTMEM16 pathway as a function of acyl chain length.** Views of the density maps for afTEM16 in C22/Ca$^{2+}$ (**A**), C18/Ca$^{2+}$ (**B**), C18/0 Ca$^{2+}$ (**C**) and C14/0 Ca$^{2+}$ (**D**) from the extracellular (top panels) and intracellular (bottom panel) side. The C1 final unsharpened maps containing nanodisc densities were aligned, resampled on the same grid, and colored according to the Z coordinate using UCSF Chimera. The density corresponding to the protein is segmented and shown in gray. Nanodisc densities are colored by displacement along the Z axis and the 0 Å reference height for the top views corresponds to H411 in the helix connecting TM5 and TM6, and Q150 on the hairpin preceding TM1 and TM2 for the bottom views. Negative values represent membrane thinning relative to the overall nanodisc. The position of the permeation pathway is denoted with arrows.

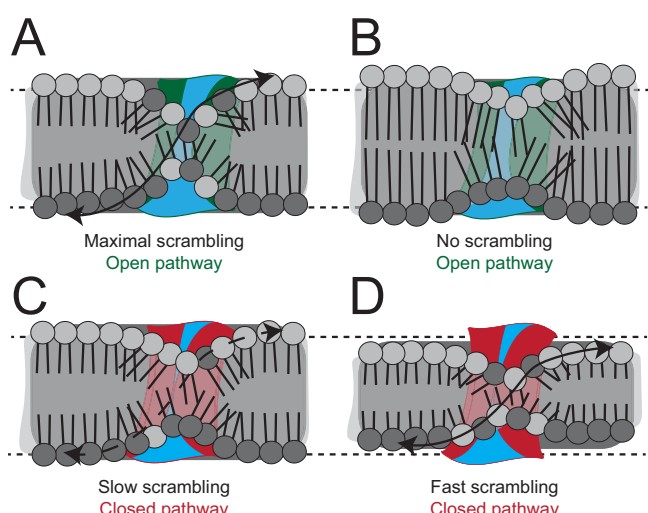

**Fig. 7 TMEM16 scramblases enable scrambling by thinning the membrane. A–D** Schematic representation of the open (**A**, **B** colored in green) and closed pathways (**C**, **D** colored in red) in membranes of different thickness. Outer and inner leaflet lipid headgroups are colored in light and dark gray, respectively. Cyan denotes regions accessible to water. Arrows denote high (solid line), low (dashed line) or no (no line) scrambling.

the lack of discrimination based on headgroup size, chemistry or charge[18,23,26,28,30,45] and scrambling of lipids conjugated to large cargoes[11]. Thus, we propose this mechanism applies to other TMEM16's.

Two lines of evidence supported the credit-card hypothesis: mutating groove-lining residues impairs lipid scrambling by

nhTMEM16 and TMEM16F[29,32] and MD simulations show lipid headgroups penetrating and traversing the whole length of the groove[25,28,29,32,46,47]. Strikingly, we find that mutating corresponding residues in afTMEM16 does not impair scrambling (Figs. 2 and 3). This contradiction could be explained if in nhTMEM16 the mutants impair scrambling by favoring groove closure rather than by impeding lipid movement through an open groove. In afTMEM16 only the Ca$^{2+}$-bound open conformation has been observed[20] (Figs. 1 and 4). In contrast, Ca$^{2+}$-bound nhTMEM16 and hTMEM16K adopt both open and closed groove conformations[22,28] and only the Ca$^{2+}$-bound closed groove conformation of mTMEM16F has been observed[21,23], suggesting in these homologs the open conformation is less stable than in afTMEM16. Several scrambling-incompetent nhTMEM16 mutants retain measurable channel activity[29], suggesting stabilization of an ion channel-like, closed-groove conformation[24]. In simulations the extracellular vestibule is embedded in the membrane and its residues interact with lipids to enable scrambling[25,28,29,32,47]. In contrast, our cryoEM maps and functional experiments suggest residues in the extracellular vestibule do not interact with lipids or play critical functional roles (Figs. 1–3). Further, our structures show several lipids with tails tightly intercalated with TM helices at the dimer cavity (Fig. 1) that might affect the dynamic rearrangements of afTMEM16 but were not considered in MD simulations. It is possible these differences in protein-membrane interface could arise from incomplete relaxation of the membrane during the initial MD equilibration steps or because of constraints imposed on the membrane by the scaffold protein in cryoEM experiments. Additional work is needed to determine the origin of this discrepancy.

The implications of our proposed membrane-thinning mechanism for scrambling (Fig. 7) can be appreciated if we

make the simplifying assumptions that (i) the energy barrier for lipid scrambling is due to the polar head and (ii) that the head can be modeled with a sphere of radius $r$ and charge $q$, then the energy barrier for scrambling, $\Delta G_{scramb}$ would be given by[48]

$$\Delta G_{scramb} = \frac{q^2}{2\varepsilon_m r} - \frac{q^2}{\varepsilon_m L} ln\left(\frac{2\varepsilon_w}{\varepsilon_m + \varepsilon_w}\right) \quad (1)$$

where $\varepsilon_{m,w}$ are the dielectric constants of the membrane and of water, respectively, and $L$ is the thickness of the membrane. Membrane thinning decreases $\Delta G_{scramb}$ as $L$ is reduced and the dielectric constant $\varepsilon_m$ is increased because of higher water access to the hydrocarbon core of the membrane[49]. Thus, when the pathway is open scrambling is fast because thinning is pronounced and the hydrophilic interior of the open groove decreases $\varepsilon_m$ (Fig. 7A). In thicker or more rigid membranes (Fig. 4)[20] scrambling is impaired because their deformation cost is higher, preventing lipids from reaching positions conducive to scrambling (Fig. 7B). When the groove is closed membrane thinning is diminished, but not absent, which allows for slow scrambling activity (Fig. 7C), that is enhanced in membranes formed from shorter chain length lipids (Fig. 7D). Notably, the proposed membrane-thinning mechanism could naturally explain how proteins lacking hydrophilic grooves, such as GPCR's and Xkr's, can scramble lipids and share functional properties with the structurally unrelated TMEM16's[10,11,38,39,41,50].

In sum, our results support a mechanism where during scrambling, lipids interact with the surface of the groove without having to penetrate and interact with its narrow and hydrophilic interior. Scrambling by the TMEM16's is modulated by two signals, binding of $Ca^{2+}$ facilitates opening of the groove while the properties of the membrane determine whether the scramblase can thin the membrane enough to enable lipid flipping. This dual control of scrambling activity, by ligand binding and membrane properties, could constitute a secondary layer of regulation that prevents undesired lipid flipping by the TMEM16's during fluctuations in cellular cytosolic $Ca^{2+}$ levels or when family members that reside in intracellular membranes are transiently localized to the plasma membrane. Similarly, rigidifying or thickening bilayer constituents, such as cholesterol, could also silence the scrambling activity of other scramblases such as GPCR's in cellular membranes.

## Methods

**afTMEM16 expression and purification**. afTMEM16 was expressed and purified following established protocols[30]. Briefly, *S. cerevisiae* transformed with the pDDGFP2 vector[51] containing the gene encoding for the afTMEM16 protein were grown in yeast synthetic drop-out medium supplemented with Uracil (CSM-URA; MP Biomedicals) and expression was induced with 2% (w/v) galactose at 30 °C for 22 h. Cells were collected, snap frozen in liquid nitrogen, lysed by cryomilling (Retsch model MM400) in liquid nitrogen (3 × 3 min, 25 Hz), and resuspended 150 mM KCl, 10% (w/v) glycerol, 50 mM Tris-HCl, pH8 supplemented with 1 mM EDTA, 5 µg ml$^{-1}$ leupeptin, 2 µg ml$^{-1}$ pepstatin, 100 µM phenylmethane sulphonylfluoride and protease inhibitor cocktail tablets (Roche). Protein was extracted using 1% (w/v) digitonin (EMD biosciences) at 4 °C for 2 h and the lysate was cleared by centrifugation at 40,000 × g for 45 min. The supernatant was supplemented with 1 mM MgCl$_2$ and 10 mM Imidazole, loaded onto a column of Ni-NTA agarose resin (Qiagen), washed with 150 mM KCl, 10% (w/v) glycerol, 50 mM Tris-HCl, pH 8 + 30 mM Imidazole and 0.12% digitonin, and eluted with 150 mM KCl, 10% (w/v) glycerol, 50 mM Tris-HCl, pH8 + 300 mM Imidazole and 0.12% digitonin. The elution was treated with Tobacco Etch Virus protease overnight to remove the His tag and then further purified on a Superdex 200 10/300 GL column equilibrated with buffer A supplemented with 0.12% digitonin (GE Lifesciences). The afTMEM16 protein peak was collected and concentrated using a 50 kDa molecular weight cut off concentrator (Amicon Ultra, Millipore).

**Liposome reconstitution**. Liposomes were prepared as described[30]. Briefly lipids in chloroform (Avanti), including 0.4% w/w tail labeled NBD-PE, were dried under N$_2$, washed with pentane and resuspended at 20 mg ml$^{-1}$ in 150 mM KCl, 50 mM HEPES pH 7.4 with 35 mM 3-[(3-cholamidopropyl)dimethylammonio]-1- propanesulfonate (CHAPS). All proteins were added at 5 µg protein/mg lipids and

detergent was removed using five changes of 150 mg ml$^{-1}$ Bio-Beads SM-2 (Bio-Rad) with rotation at 4 °C. Calcium or EGTA were introduced using sonicate, freeze-thaw cycles. Chain length experiments were done in the background of 7PC:3PG due to the availability of the long chain lipids. Lipids used were: 1,2-dimyristoyl-sn-glycero-3-phosphocholine (DMPC, 14:0), 1,2-dimyristoyl-sn-glycero-3-phospho-(1′-rac-glycerol) (DMPG, 14:0), 1,2-dipalmitoleoyl-sn-glycero-3-phosphocholine (16:1), 1,2-dipalmitoleoyl-sn-glycero-3-(1′-rac-glycerol) (16:1) 1-palmitoyl-2-oleoyl-glycero-3-phosphocholine (POPC, 16:0-18:1), POPG (16:0-18:1), 1,2-dioleoyl-sn-glycero-3-phosphocholine (DOPC, 18:1), 1,2-dioleoyl-sn-glycero-3-phospho-(1′-rac-glycerol) (DOPG, 18:1), 1,2-dieicosenoyl-sn-glycero-3-phosphocholine (20:1), 1,2-dieicosenoyl-sn-glycero-3-(1′-rac-glycerol) (20:1), 1,2-dierucoyl-sn-glycero-3-phosphocholine (DEPC, 22:1) and 1,2- dierucoyl-phosphatidylglycerol (DEPG, 22:1).

Reconstitution efficiency of WT and mutant afTMEM16 in each condition was comparable, as estimated by SDS-PAGE electrophoresis of samples collected before and at the end of reconstitution procedure[29].

**Scrambling assay**. The scrambling assay was carried out as described[11]. Briefly, in protein free liposomes, fluorescently labeled lipids evenly distribute between leaflets and addition of dithionite causes a ~50% decrease of fluorescence. In liposomes containing an active afTMEM16 scramblase, dithionite addition results in a nearly complete loss in fluorescence. Since not all liposomes contain an active scramblase and due to the presence of multilamellar liposomes, a fraction of NBD fluorophores is shielded from the dithionite, resulting in incomplete fluorescence loss. Liposomes were extruded through a 400 nm membrane and 20 µl were added to a final volume of 2 mL of 50 mM HEPES pH 7.4, 300 mL KCl + 0.5 mM Ca(NO$_3$)$_2$ or 2 mM EGTA. The fluorescence intensity of the NBD (excitation-470 nm emission-530 nm) was monitored over time with mixing in a PTI spectrophotometer and after 100 s sodium dithionite was added at a final concentration of 40 mM. Data was collected using the FelixGX 4.1.0 software at a sampling rate of 3 Hz.

**Quantification of scrambling activity**. Quantification of the scrambling rate constants by afTMEM16 was determined as described[11,29]. Briefly, the fluorescence time course was fit to the following equation

$$F_{tot}(t) = f_0\left(L_i^{PF} + (1 - L_i^{PF})e^{-\gamma t}\right) + \frac{(1 - f_0)}{D(\alpha + \beta)} \quad (2)$$
$$\left\{\alpha(\lambda_2 + \gamma)(\lambda_1 + \alpha + \beta)e^{\lambda_1 t} + \lambda_1\beta(\lambda_2 + \alpha + \beta + \gamma)e^{\lambda_2 t}\right\}$$

where $F_{tot}(t)$ is the total fluorescence at time $t$, $L_i^{PF}$ is the fraction of NBD-labeled lipids in the inner leaflet of protein free liposomes, where $\gamma$ is the rate constant of dithionite reduction, $f_0$ is the fraction of protein-free liposomes in the sample, $\alpha$ and $\beta$ are respectively the forward and backward scrambling rate constants and

$$\lambda_1 = -\frac{(\alpha + \beta + \gamma) - \sqrt{(\alpha + \beta + \gamma)^2 - 4\alpha\gamma}}{2},$$
$$\lambda_2 = -\frac{(\alpha + \beta + \gamma) + \sqrt{(\alpha + \beta + \gamma)^2 - 4\alpha\gamma}}{2}$$
$$D = (\lambda_1 + \alpha)(\lambda_2 + \beta + \gamma) - \alpha\beta$$

The free parameters of the fit are $f_0$, $\alpha$ and $\beta$ while $L_i^{PF}$ and $\gamma$ are determined from experiments on protein-free liposomes. In protein-free vesicles a very slow fluorescence decay is visible, likely reflecting a slow leakage of dithionite into the vesicles or the spontaneous flipping of the NBD-labeled lipids. A linear fit was used to estimate that the rate of this process is $L = (5.4 \pm 1.6)10^{-5} \text{s}^{-1}$ ($n > 160$)[29]. For wiltype and mutant afTMEM16 the leak is >2 orders of magnitude smaller than the rate constant of protein-mediated scrambling and therefore is negligible. All conditions were tested side by side with a control preparation of WT afTMEM16 reconstituted in C18 lipids. In some rare cases this control sample behaved anomalously, judged by scrambling fit parameters outside 3 times the standard deviation of the mean for the wildtype. In these cases, the whole batch of experiments was disregarded.

**MSP1E3/MSP2N2 purification and nanodisc reconstitution**. MSP1E3 and MSP2N2 was expressed and purified as described[52]. Briefly, MSP1E3 in a pET vector (Addgene #20064) was transformed into the BL21-Gold (DE3) strain (Stratagene). Transformed cells were grown in LB media supplemented with Kanamycin (50 mg l$^{-1}$) to an OD$_{600}$ of 0.8 and expression was induced with 1 mM IPTG for 3 h. Cells were harvested and resuspended in 40 mM Tris-HCl pH 78.0, 300 mM NaCl supplemented with 1% Triton X-100, 5 µg ml$^{-1}$ leupeptin, 2 µg ml$^{-1}$ pepstatin, 100 µM phenylmethane sulphonylfluoride and protease inhibitor cocktail tablets (Roche). Cells were lysed by sonication and the lysate was cleared by centrifugation at 30,000 × g for 45 min at 4 °C. The lysate was incubated with Ni-NTA agarose resin for 1 h at 4 °C followed by sequential washes with: 40 mM Tris-HCl pH 78.0, 300 mM NaCl +1% triton-100, +50 mM sodium cholate, +20 mM imidazole and +50 mM imidazole. The protein was eluted with buffer C +400 mM imidazole, desalted using a PD-10 desalting column (GE life science) equilibrated with 150 mM KCl, 50 mM Tris pH 8.0 supplemented with 0.5 mM EDTA. The

**Table 2 Electron microscopy data collection parameters.**

| C18/Ca²⁺ | | C18/0 Ca²⁺ | C14/0 Ca²⁺ | C22/Ca²⁺ (MSP1E3) | C22/Ca²⁺ (MSP2N2) | D511A/E514A C14/Ca²⁺ |
|---|---|---|---|---|---|---|
| Microscope/camera | Krios/K3 | Krios/K3 | Krios/K2 | Krios/K3 | Krios/K3 | Krios/k3 |
| Acquisition | SerialEM | Leginon | Leginon | SerialEM | SerialEM | SerialEM |
| Accelerating Voltage (kV) | 300 | 300 | 300 | 300 | 300 | 300 |
| Number of frames | 30 | 40 | 50 | 30 | 30 | 30 |
| Dose (e-/ Å²) | 42.8 | 59.1 | 71.7 | 42.8 | 44.4 | 42.2 |
| Defocus range | 0.5-2.3 | 1.5-2.0 | 1.5 -2.3 | 0.5-2.3 | 0.5-2.3 | 0.5-2.3 |
| Exposure time | 2.7 | 2.8 | 10 | 2.4 | 2.7 | 1.7 |
| Pixel size | 0.53 (super resolution) | 0.426 (super resolution) | 1.06 (counting) | 0.53 (super resolution) | 0.53 (super resolution) | 0.53 (super resolution) |
| Energy filter | Yes (20 eV) | N/A | Yes (20 eV) | Yes (20 eV) | Yes (20 eV) | Yes (20 eV) |

final protein was concentrated to ~8 mg ml⁻¹ using a 30 kDa molecular weight cut off concentrator (Amicon Ultra, Millipore), flash frozen and stored at −80 °C.

Reconstitution of afTMEM16 in nanodiscs was carried out as described[20]. Lipids in chloroform (Avanti) were dried under N₂, washed with pentane and resuspended in buffer D and 40 mM sodium cholate (Anatrace) at a final concentration of 20 mM. Molar ratios 1:0.8:50 MSP1E3:afTMEM16:lipids and 1:0.8:140 for MSP2N2:afTMEM16:lipids were mixed at a final lipid concentration of 7 mM and incubated at room temperature for 20 min. Detergent removal was carried out at 4 °C via incubation with Bio-Beads SM-2 (Bio-Rad) with agitation for 2 h and then overnight with fresh Bio-Beads SM2 at a concentration of 200 mg ml⁻¹. The reconstitution mixture was purified using a Superose6 Increase 10/300 GL column (GE Lifesciences) pre-equilibrated with 50 mM HEPES pH 8.0 150 mM KCl plus 5 mM EDTA or 0.5 mM CaCl₂ and the peak corresponding to afTMEM16-containing nanodiscs was collected for cryo electron microscopy analysis.

**Atomic force microscopy**

*Sample preparation.* Large unilamellar vesicles (LUVs) with concentration 0.01 mg/ml were deposited on freshly cleaved mica for 20 min, followed by careful rinsing with imaging buffer (50 mM HEPES, pH 7.4, 300 mM KCl). The sample was kept in imaging buffer for all AFM measurements.

*Imaging and mechanical measurements.* All AFM measurements were performed using a JPK Nanowizard 4 AFM (Bruker, Berlin, Germany). Images were acquired in tapping mode by using cantilevers (FastScan D, Bruker) with a normal spring constant of ~0.25 N/m, and a resonant frequency of ~110 kHZ in liquid. The sensitivity of cantilevers was measured on mica and the spring constant were calibrated by using thermal tune methods[53]. Force-distance curves were acquired at 500 nm/s.

*Data processing.* All the data analysis were performed using JPK data processing software. The Young's modulus of the supported lipid bilayers was derived by fitting the force distance curves with the following model[54,55],

$$F = \frac{E}{1-\nu^2} \frac{2\tan\alpha}{\pi} \delta^2 \quad (3)$$

where $F$ is the Force, $E$ is the Young's modulus, $n$ is the Poisson's ratio, $d$ the indentation (vertical tip position), and $\alpha$ is the half cone angle or half face angle of the pyramidal tip shape.

The area stretch modulus ($k_A$) and bending stiffness ($k_c$) of the supported lipid bilayers were calculated based on the measured Young's modulus ($E$) and membrane thickness ($h$) using the following equations (the Poisson's ratio $n$ is assumed 0.5[56]):

$$k_A = \frac{Eh}{1-\nu^2} \quad (4)$$

$$k_c = \frac{Eh^3}{24(1-\nu^2)} \quad (5)$$

**Grid preparation**. 3.5–5 μL of afTMEM16-containing nanodiscs (2–7 mg mL⁻¹) supplemented with 1.5 or 3 mM Fos-Choline-8-Fluorinated (Anatrace) was applied to a glow-discharged UltrAuFoil R1.2/1.3 300-mesh gold grid (Quantifoil) and incubated for one minute under 100% humidity at 15 °C. Following incubation, grids were blotted for 2 s and plunge frozen in liquid ethane using a Vitrobot Mark IV (FEI).

**Image processing**. Data collection was carried out for each structure using the parameters specified in Table 2. Image analysis was carried out using Relion 3.0 or 3.1 beta[57]. Motion correction was carried out using the Relion implementation of MotionCorr2[57] and contrast transfer function (CTF) estimation was performed

using CTFFIND4[58] via Relion. For all datasets except the wildtype afTMEM16 in 50% 14:0 lipids, 2× binning was used for motion correction. All refinements were run in Relion initially without a mask, and the converged refinement was continued with a mask excluding the nanodisc or micelle. The final resolution of all maps was determined by applying a soft mask around the protein and the gold-standard Fourier shell correlation (FSC) = 0.143 criterion using Relion Post-Processing. Relion was used to estimate the local resolution for all final maps. Processing strategy for the various datasets differed; details are described below.

*Processing of dataset for afTMEM16 in 50% C14 lipids in 0 Ca²⁺.* 7432 micrographs were collected and following manual inspection, 5253 were included for analysis with Relion 3.0. Following motion correction and CTF estimation, auto-picking was carried out using a 3D volume of the afTMEM16/nanodisc complex low pass filtered to 30 Å and yielded 1,728,771 particles. Particles were extracted using a box size of 271 Å with 2× binning and subjected to two rounds of 2D classification. 739,898 particles with structural features resembling the afTMEM16-nanodisc complex were selected and subjected to 3D classification without symmetry. 567,785 particles from well-defined 3D classes were extracted without binning and subjected 3D auto-refinement, which resulted in a 3.7 Å reconstruction. Several rounds of CTF refinement[57] and Bayesian polishing[59] followed by refinement and 3D classification with and without particle alignment were carried out, in which particles not resembling afTMEM16/nanodisc complex were discarded, resulting in a subset of 334,014 particles. This subset was used for extensive classification to identify alternate conformations described below. After no different conformations were identified, additional rounds of 3D classification without alignment and a mask excluding the nanodisc were used to improve the resolution, resulting in a final subset of 244,507 particles. Additional rounds of CTF refinement and Bayesian polishing led to a masked reconstruction with a resolution 3.4 Å, which was C2 symmetric. This initial map (without application of C2 symmetry) was used to analyze the effects on the membrane. Additional rounds of refinement were carried out with C2 symmetry applied resulting in a final reconstruction of 3.3 Å which was used for model building.

*Processing of dataset for afTMEM16 in C22 lipids in MSP1E3 nanodiscs in 0.5 mM Ca²⁺.* 9395 micrographs were collected and after manual inspection 7157 were included for analysis in Relion 3.0 and 3.1 beta. Following motion correction with 2× binning (pixel size 1.06) and CTF estimation, auto-picking was carried out using a 3D volume of TMEM16/nanodisc complex low pass filtered to 30 Å and picked 2,643,590 particles. These particles were extracted using a box size of 271 Å with 4× binning (pixel size 2.12) and subjected to two rounds of 2D classification. 1,923,936 particles with structural features resembling the afTMEM16-nanodisc complex were selected and subjected to 3D classification without symmetry. 1,094,546 particles from well-defined 3D classes were extracted with 2× binning (pixel size 1.06) and subjected to several rounds of CTF refinement[57] and Bayesian polishing[59] followed by refinement and 3D classification with and without particle alignment, in which particles not resembling afTMEM16/nanodisc complex were discarded, resulting in a subset of 538,468 particles. This subset was used for extensive classification to identify alternate conformations described below, with the exception that classification on monomers was carried out on the reduced subset of 245,824 particles (see below). No attempts to find alternate conformations were successful. To improve the resolution, several additional rounds of 3D classification without alignment and CTF refinement and Bayesian polishing were carried out, leading to a subset of 245,824 particles. Using the Bayesian polishing job in Relion 3.1, these particles were further un-binned to a pixel size of 0.7066 (~1.3× binning from the micrographs). After additional rounds of 3D classification without alignment and CTF refinement and Bayesian polishing, a final subset of 132,332 particles were selected which resulted in a 2.76 Å reconstruction. The protein was not centered in the nanodisc and density for TM6 was missing from one monomer as previously described[20]. Signal subtraction of the nanodisc density resulted a symmetric reconstruction and additional rounds of refinements with

imposed C2 symmetry were carried out, resulting in a final 2.7 Å reconstruction, which was used for model building. The C1 reconstruction prior to signal subtraction was used to analyze the effects on the membrane. Signal subtraction and symmetry expansion were also carried out on the 2× binned particles (pixel size 1.06) and the described classification of the monomers (below) was carried out on these particles.

*Processing of dataset for afTMEM16 in C22 lipids in MSP2N2 nanodiscs in 0.5 $Ca^{2+}$.* 7915 micrographs were collected and after manual inspection 5805 were included for analysis in Relion3.0 and cryoSPARC[57,60]. Following motion correction with 2× binning (pixel size 1.06) and CTF estimation, Laplacian of gaussian auto-picking with conservative parameters was carried out[57]. 447,947 particles were extracted with a box size of 271 Å with 2× binning (pixel size 1.06) and subjected to two rounds of 2D classification to generate templates for auto-picking. 1,641,081 particles picked with Template-based auto-picking were extracted with a box size of 305 Å and 4× binning (pixel size 2.12) and subjected to three rounds of 2D classification. 1,168,291 particles with features of the afTMME16/nanodisc complex were extracted with a box size of 305 Å and 2× binning (pixel size 1.06) and were imported into cryoSPARC and subjected to heterogenous refinement with forced classification with one model resembling the afTMEM16/nanodisc complex and two models resembling empty nanodiscs[60]. 208,670 particles from the class resembling the afTMEM16/nanodisc complex were subjected to several rounds of Ab-initio reconstruction with two classes in which particles that did not lead to models resembling the afTMEM16/nanodisc complex were discarded. A final subset of 31,890 particles resulted in a 4 Å reconstruction via non-uniform refinement. These particles were brought to Relion 3.0 using the csparc2star.py script[61] and subjected to auto-refinement with local angular searches without symmetry. After three rounds of CTF refinement[57] and Bayesian polishing[59], these particles refined to 3.7 Å. As in the structure of afTMEM16 in 22:1 lipids in MSP1E3 nanodiscs, the protein was not centered in the nanodisc and density for TM6 was missing from one monomer. Signal subtraction of the nanodisc density was performed in Relion and the subtracted particles were imported into cryoSPARC and subjected to non-uniform refinement with C2 symmetry enforced, resulting in a 3.5 Å reconstruction, which was used for model building. The C1 reconstruction prior to signal subtraction was used to analyze the effects on the membrane.

*Processing of dataset for afTMEM16 in C18 lipids in MSP1E3 nanodiscs in 0.5 mM $Ca^{2+}$.* 8891 micrographs were collected and after manual inspection 6336 were included for analysis in Relion 3.0 and 3.1 beta. Following motion correction with 2× binning (pixel size 1.06) and CTF estimation, auto-picking was carried out using a 3D volume of the afTMEM16/nanodisc complex low pass filtered to 30 Å and picked 4,551,732 particles. These particles were extracted with a box size of 271 Å with 4× binning (pixel size 2.12) and subjected to two rounds of 2D classification from which 2,645,466 particles displayed afTMEM16/nanodisc complex features and were subjected to two rounds of 3D classification. 1,601,172 particles from classes resembling the afTMEM16/nanodisc complex were extracted with 2× binning (pixel size 1.06) and subjected an additional round of 3D classification. 1,335,468 particles were subjected to auto-refinement and yielded a 2.8 Å reconstruction. Following three rounds of CTF refinement[57] and Bayesian polishing[59], these particles were subjected to extensive classification to identify other conformations (see below). No attempts to identify alternate protein conformations were successful. Using the Bayesian polishing job in Relion 3.1, these particles were further un-binned to a pixel size of 0.7066 (*~1.3× binning from the micrographs). After three rounds of CTF refinement[57] and Bayesian polishing[59], these particles yielded a C2 symmetric 2.5 Å reconstruction. Additional rounds of refinement and classification without alignment were performed with C2 symmetry enforced, resulting in a final subset of 994,187 particles and a Å 2.28 reconstruction. To identify additional associated lipids, the 1,335,468 particles unbinned to 0.7066 Å/pix were symmetry expanded and further classified with a mask on one monomer, with and without local alignment with varying T values. One class with 1,058,829 monomers showed strong density for several additional lipids surrounding the permeation pathway. Local refinement of these particles yielded a Å 2.28 reconstruction which was used to build additional lipids (Supplementary Figs. 1 and 2).

*Processing of dataset for afTMEM16 D511A/E514A in C14 lipids in 0.5 mM $Ca^{2+}$.* 9853 micrographs were collected and following manual inspection, 7693 were included for analysis with Relion 3.0. Following motion correction and CTF estimation, 3136 particles were manually picked to generate 2D class averages that were subsequently used as templates to pick a total of 3,820,960 particles. Particles were extracted using a box size of 271 Å with 4× (pixel 2.12) binning and subjected to two rounds of 2D classification. 2,151,574 particles with structural features resembling the afTMEM16-nanodisc complex were selected and subjected to 3D classification without symmetry. 886,967 particles from well-defined 3D classes were extracted with 2× binning (pixel size 1.06) and subjected 3D auto-refinement, which resulted in a 3.56 Å reconstruction. CTF refinement[57] and Bayesian polishing[59] followed by refinement and 3D classification with and without particle alignment were carried out, in which particles not resembling afTMEM16/nanodisc complex were discarded, resulting in a subset of 483,023 particles. This subset was used for extensive classification to identify alternate conformations described below. After no alternate conformations were identified, additional

rounds of 3D classification without alignment and a mask excluding the nanodisc were used to improve the resolution, resulting in a final subset of 155,902 particles. Additional rounds of CTF refinement[57] and Bayesian polishing[59] led to a masked reconstruction with a resolution 3.3 Å, which was C2 symmetric. This initial map (without application of C2 symmetry) was used to analyze the effects on the membrane. Additional rounds of refinement were carried out with C2 symmetry applied resulting in a final reconstruction of 3.08 Å which was used for model building.

*Processing of dataset for afTMEM16 in C18 lipids in 0 $Ca^{2+}$.* 6400 micrographs were collected and included for analysis with Relion 3.0. Following motion correction and CTF estimation, 5315 particles were manually picked to generate 2D class averages that were subsequently used as templates to automatically pick a total of 2,411,226 particles. Particles were extracted using a box size of 218 Å with 4× binning (pixel size 2.12) and subjected to 2D classification. 2,142,631 particles were selected and extracted with 2× binning (pixel size 0.85) for another round of 2D classification. 1,931,850 particles with structural features resembling the afTMEM16-nanodisc complex were selected and subjected to 3D classification without symmetry. 525,407 particles from well-defined classes were subjected 3D auto-refinement, which resulted in a 3.46 Å reconstruction. CTF refinement[57] and Bayesian polishing[59] followed by refinement and 3D classification with and without particle alignment were carried out, in which particles not resembling afTMEM16/nanodisc complex were discarded, resulting in a subset of 461,108 particles. This subset was used for extensive classification to identify alternate conformations described below. After no alternate conformations were identified, additional rounds of 3D classification without alignment and a mask excluding the nanodisc were used to improve the resolution, resulting in a final subset of 151,197 particles. Additional rounds of CTF refinement[57] and Bayesian polishing[59] led to a masked reconstruction with a resolution 3.25 Å, which was C2 symmetric. This initial map (without application of C2 symmetry) was used to analyze the effects on the membrane. Additional rounds of refinement were carried out with C2 symmetry applied resulting in a final reconstruction of 3.07 Å which was used for model building.

*Classification to identify alternate conformations.* On all datasets except for afTMEM16 in C22/$Ca^{2+}$ in MSP2N2 nanodiscs, we carried out extensive 3D classification to identify potential alternate conformations. We tried classification with alignment with global searches and with local searches both with a without a mask excluding the nanodisc. The following approaches were tried: (i) varying the starting model (open and closed permeation pathways), the low pass filter (10–20 Å), the T parameter (4–10), and number of classes (6–20); (ii) 3D classification without alignment (using the angles from refinement) with masks excluding the nanodisc and varying the same parameters as classification with alignment; (iii) symmetry expansion and signal subtraction to isolate the monomers followed by subsequent focused classification/refinement on the permeation pathway (TM3–7)[62] to account for potential alternate conformations between two protomers; (iv) focused classification on the monomers on the expanded but not subtracted particles; (v) refinement using cryoSPARC[60] for all described structures and cisTEM[63] for the C14/0 $Ca^{2+}$ dataset using particles picked in Relion. In cryoSPARC, we sorted using one round of 2D classification followed by several rounds of ab initio model generation using 3–5 classes, discarding particles from each round that did not result in a reasonable reconstruction. Heterogeneous refinement with 3–5 classes was carried out following each round of ab initio model generation. We also tried 3D variability analysis on all particles selected from 2D classification and on a reduced set of particles sorted using ab initio and heterogenous refinement, both of which revealed a single population and no movements of the protein. In cisTEM we imported particles picked in Relion and sorted using 2D classification and tried global and local refinement using 5–10 classes. None of these approaches led to the observation of a class with an open permeation pathway.

**Model building and refinement.** Previous apo or $Ca^{2+}$-bound afTMEM16 structures were used as a starting model and fit into the new maps using several rounds of PHENIX real space refinement[64] including morphing and simulated annealing every macrocycle. The improved resolution in these structures, particularly the C18/$Ca^{2+}$ allowed us to correct errors in the earlier models and to build additional parts of the protein including the hairpin preceding TM3 and the extracellular region between TM5 and TM6 (Fig. 1A, B). Lipids were modeled initially as POPG (PGW) and truncated according to the observed density. In most cases the full headgroup could not be resolved so the lipids were truncated at the phosphate atom. The models were inspected and areas that were better resolved were build de novo. The model was improved iteratively by real space refinement in PHENIX imposing non-crystallographic symmetry restraints and secondary structure restraints followed by manual inspection and removal of outliers. For all models, the unsharpened maps were used to aid in building. Residue ranges and positions with truncated side chains are listed in Supplementary Table 1.

**Model validation.** To validate the refinement, the FSC between the refined model and the final map was calculated (FSCsum). To evaluate for over-fitting, random

shifts of up to 0.3 Å were introduced in the final model and the modified model was refined using PHENIX[64] against one of the two unfiltered half maps. The FSC between this modified-refined model and the half map used in refinement (FSCwork) was determined and compared to the FSC between the modified-refined model and the other half map (FSCfree) which was not used in validation or refinement. The similarity in these curves indicates that the model was not over-fit. The quality of all three models was assessed using MolProbity[65] which indicates that the models are of high quality. Statistics are found in Supplementary Table 2.

**Statistics and reproducibility.** For each cryoEM reconstruction, at least 1,000,000 individual particles were picked from cryoEM images and used for structural analysis following standard protocols. Data statistics and evaluation of resolution are documented in Supplementary Table 2. Functional experiments were repeated 6+ times from 2+ independent preparations and reconstitutions.

**Reporting summary.** Further information on research design is available in the Nature Research Reporting Summary linked to this article.

## Data availability

The data that support this study are available from the corresponding author upon reasonable request. All models and associated cryoEM maps have been deposited into the Electron Microscopy Data Bank (EMDB) and the Protein Data Bank (PDB). WT afTMEM16 dimer in C18 lipids and MSP1E3 nanodiscs in the presence of Ca:²⁺ 7RXH and EMD-24730; signal subtracted monomer: 7RXG and EMD-24731; WT afTMEM16 in the absence of Ca:²⁺ 7RXB and EMD-24727; WT afTMEM16 in C22 lipids in the presence of Ca²⁺ and MSP1E3 nanodiscs: 7RX2 and EMD-24722; and MSP2N2 nanodiscs: 7RWJ and EMD-24717; WT afTMEM16 dimer in C14 lipids and MSP1E3 nanodiscs in the absence of Ca:²⁺ 7RX3 and EMD-24723; D511A/E514A afTMEM16 dimer in C14 lipids and MSP1E3 nanodiscs in the absence of Ca:²⁺ 7RXA PDB and EMD-24726. Source Data are provided with this paper.

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

## Acknowledgements

The authors thank members of the Accardi lab and Richard Hite for helpful discussions. We wish to thank reviewers of Biophysics Colab, A. Ballesteros, V. Kalienkova and K. Swartz, as well as the anonymous reviewers of Nature Communications for their comments. The work was supported by National Institutes of Health (NIH) Grant R01GM106717 (to A.A.), by a Margaret and Herman Sokol Fellowship from Weill Cornell Medicine (M.E.F.), by the KBRI Basic Research Program through Korea Brain Research Institute funded by Ministry of Science and ICT (21-BR-01-08 to B.-C.L.) and National Research Foundation of Korea (NRF) grant funded by the Korea government (MSIT) (2019R1C1C1002699 to B.-C.L.). Some of this work was performed at the Simons Electron Microscopy Center and National Resource for Automated Molecular Microscopy located at the New York Structural Biology Center, supported by grants from the Simons Foundation (SF349247), NYSTAR, and the NIH National Institute of General Medical Sciences (GM103310). Part of the work was performed at the Cryo-EM Core Facility at University of Massachusetts Chan Medical School with the help of Dr. Kangkang Song and Dr. Chen Xu. Initial screening was performed at NYU Langone Health's Cryo–Electron Microscopy Laboratory (RRID: SCR_019202).

## Author contributions

M.F., Z.F. and A.A. designed the experiments; M.F., Z.F., O.E.A., Y.P. and B.-C.L. performed experiments; M.F., Z.F., O.E.A., Y.P., B.-C.L., X.C., E.F., S.S. and A.A. analyzed the data; M.F., S.S. and A.A. wrote the paper. All authors edited the manuscript.

## Competing interests

The authors declare no competing interests.
