## [Peer Review File · Nature Communications]

REVIEWER COMMENTS

Reviewer #1 (Remarks to the Author):

This is an important contribution from the Accardi lab on the mechanisms of lipid scrambling by the TMEM16 proteins. First, they solved the structure of aTMEM16 in nanodiscs at a significantly improved resolution that provides significant insights in the structure of the protein in a lipid environment. The coordination of lipids at the dimer interface and in the vicinity of the scrambling pathway are described in detail. These images reveal considerable detail about how the aTMEM16 thins the membrane near the scrambling pathway and lowers the energy barrier for lipid translocation between leaflets. An exciting observation is the discovery that mutation of residues that interact with the lipid headgroups in the pathway or the extracellular vestibule do not affect scrambling significantly. In support of the membrane thinning model of scrambling, scrambling is dependent on acyl chain length and membrane thickness. Remarkably, the structures of the scrambling pathway do not correlate with scrambling activity and aTMEM16 has an open groove in C22 lipids that do not scramble and scrambling in zero Ca which occurs with C14 lipids. The authors take this argument further by showing that the D511A/E514A mutant which does not bind Ca and adopts the closed conformation still scrambles lipids depending on acyl chain length. These findings are very inconsistent with the credit-card mechanism of lipid scrambling that has dominated the field over the past decade and explains several aspects of scrambling that had been puzzling. This is a very significant contribution

Comments:

While it might seem obvious, the authors should discuss why scrambling does not take place at the dimer interface.

The number of experiments should be noted in each legend.

Page 4. The authors should also mention that at least two MD simulation papers (Jiang et al., and Bethel et al.) have suggested that membrane thinning contributes to scrambling.

Page 5. "3 1-palmitoyl-2-oleoyl-sn-glycero-3-phosphoethanolamine (POPE): 1 1-palmitoyl-2-oleoyl-sn-glycero-3-phospho-(1'-racglycerol) POPG nanodiscs 15, Ca rmsd ~0.8 Å." The sentence is confusing to read. First, better to say 3:1 POPE (...): POPG (...) and make the Ca start a new sentence.

Figure 1A – please label helices.

Figure 1 C,D,E would benefit from some more labels identifying helices. I am having trouble understanding Fig. 1E because it seems inconsistent with the statement that the extracellular vestibule does not directly interact with lipids.

Page 11. I don't understand the sentence "This suggests lipid headgroups only need to traverse the wide intracellular vestibule of the pathway, below the constriction formed by T325 and Y432 (Fig. 2A)." The word traverse implies the conclusion that P3 has moved from the intracellular to the extracellular leaflet, but there is no data to support this. Also, the figure does not show Y432

Page 12. "synergistically lower the energy barrier for scrambling" grammar.

Page 13. Abbreviations for POPC and POPG (PC, PG?). Also, there are two ratios here: acyl

chain saturation and lipid ratio. So, the phrase "mix of 14:1 lipids" should be clarified.

Figure 4. The groove is open in panel D and closed in panel E. For those who are not cognoscenti of the TMEM16 proteins, the authors should describe these structures more explicitly. Maybe add a cartoon to this figure?

Page 18. The authors do not define MSP1E3 and MSP2N2 nanodiscs in the text – they are only defined in Suppl. Fig. 5.

Page 21. The authors state that there is a discrepancy between the membrane thinning model and previously published MD simulations that show lipids traversing the groove. However, it seems to me that the membrane thinning model does not exclude the possibility that lipids move through the groove - at least some of the time. The groove may not be a specific pathway, but one that has a low energy barrier. I think the authors should not belabor the equilibration issue.

Reviewer #2 (Remarks to the Author):

This work is a significant advance in the mechanistic understanding of TMEM16 lipid scramblase. The proposed membrane thinning mechanism of scrambling is an intriguing and elegant solution that has escaped experimental validation until now. The combination of structural and functional studies to directly visualize conformational changes and measure scrambling rates with native and mutant protein provides powerful evidence for the proposed mechanism. These results will have a broad impact on the field and provides a foundation for studying other members of the TMEM scramblases and publication in Nature Communications is strongly recommended after minor revisions to improve the presentation and clarify several points.

Below are listed specific strengths and some minor weaknesses combined with presentation comments regarding the current submission

Major Strengths

1. Provides a clear mechanistic understanding of the TMEM16 lipid scramblase.
2. Alters the landscape of our understanding of how TMEM16 lipid scramblase function.
3. The cryoEM studies are of excellent quality and combine the use of lipid nano-discs with lipids of varying chain lengths to observe conformational changes.
4. The scramblase assays provide a key functional linkage between the structure and the thinning mechanism and greatly strengthen the study.
5. Provides a foundation for future studies of other TMEM scramblases

Minor weaknesses and presentation comments

1. The comparative scrambling rates shown in figure 3 should include at a minimum a statistical t-test between WT and each respective mutant. The statement that the mutations have no effect is somewhat misleading as several of the mutants would appear to have a statistically significant difference at 0 M Ca⁺⁺.
2. Fig 4A should use nomenclature D511A D514E and not DA/DE
3. Fig 6 is not very informative in the present format. Firstly, the zero height position should be stated in relation to residue within the structure. Secondly, this would be more impactful to the reader if this were presented in the plane of the membrane and cut at the single lipid layer and magnified to show the permeation pathway. Or at least in close up of a

single permeation pathway. This reader would prefer both.

4. Fig 7 would be enhanced by coloring the two leaflet headgroups different to accentuate the scrambling, i.e. inner leaflet to outer one color and visa versa.

5. The EM data table S1 should list defocus as a positive number by convention. Listing the dose and exposure times to 4 significant figures is both unwarranted and unrealistic that they have been determined to such precision.

6. EM data table S4 again uses unrealistic precision.

7. The EM data table S4 does not list B factors for the protein or lipid. While the lipid densities shown in the supplement are convincing, I believe it is important to list at least the average B factor for the protein and the lipids in this table.

8. There are a small number of typos that should be corrected.

Reviewer #3 (Remarks to the Author):

This study examines afTMEM16, a fungal calcium-activated scramblase, and compares its cryo-EM structure in nanodiscs with its ability to scramble lipids in proteoliposomes composed of lipids with different fatty acid chain lengths. The abstract summarizes their findings with statements "...cryoEM structure...showing how rearrangement of individual lipids at the open pathway results in pronounced membrane thinning" and "our results show how afTMEM16 thins the membrane to enable scrambling and that an open hydrophilic pathway is not a structural requirement to allow rapid transbilayer movement of lipids." The findings that scramblase activity does not depend on the extracellular groove-lining residues or the open conformation of the groove are very interesting and significant. It would be important to provide reviewers with the density maps of the calcium bound afTMEM16 and address the following questions:

1. Please measure the membrane thickness at the dimer interface and also at the lipid pathway, in addition to the distance between the phosphate atoms of the headgroups of P3 and P4, in nanodiscs with C18 lipids, with or without calcium. Given that the lack of support for a conveyor belt mechanism based on their mutagenesis results, it is important to document the membrane thickness around afTMEM16.

2. The authors found no effects of mutagenesis aimed at disrupting the headgroup interactions of P1-P2, P4-P5-P6 or P2-P5-P6. What about mutagenesis aimed at disrupting the headgroup interactions of P3?

3. Please provide measurements of the membrane thickness around afTMEM16 in nanodiscs with C22 lipids, and D511A/E514A afTMEM16 in nanodiscs with C14 lipids.

4. The nanodisc densities in Fig. 6 appears to show thinning at the rim of nanodiscs. For the main theme of this paper, it is important to quantify the membrane thickness in the vicinity of afTMEM16 beyond P3 and P4.

5. Whereas membrane distortion could tilt lipids towards the protein rather than the surface of bilayer, it is important for the authors to demonstrate membrane thinning that facilitates lipid scrambling and make the point clearly in discussion.

6. On page 14-15, "despite the high resolution of the C22/Ca²⁺ MSP1E3 map, we detect only weak signals for lipids associated with the pathway-delimiting helices TM4 and TM6." On page 17-18, "No lipids could be resolved near the closed pathway in these structures. The average resolution of these datasets is lower than that of the two Ca²⁺-bound structures, preventing us from drawing mechanistic inferences from this observation. " However, the resolution of the structure is 2.3 Å in C18, 2.7 Å in C22 and around 3.1 Å for the Ca-free ones. Compare to 2.7 Å, 2.3 Å is significant higher in resolution. It is hard to make such conclusion even between the two Ca²⁺ bound structures. It would help if the authors can share the density maps of the two Ca²⁺ structures, so we can assess how much improvement these maps have made as compared to the published ones with the

same condition and how reliable the comparison of lipids is between the two maps.

Reviewer #4 (Remarks to the Author):

The manuscript by Falzone et al. is intended to describe the mechanism of lipid scrambling by TMEM 16 scramblases. More specifically, the authors try to solve the paradox that, according to the currently prevailing 'credit-card' mechanism, with the headgroups forming specific interactions with polar and charged residues lining the protein hydrophilic groove, scramblases should discriminate among lipids based on their headgroups but not their tails. However, several scramblases fail to show selectivity among lipids with different headgroups, while others lack clear hydrophilic grooves. The authors propose instead that, at least with afTMEM16, the mechanism is based on the protein thinning the membrane to enable scrambling. They further contend that an open hydrophilic pathway is not a structural requirement to allow rapid transbilayer movement of lipids.

They have studied a fungal TMEM 16 reconstituted in nanodiscs, using mainly cryo EM. They have achieved a 2.3 Å resolution structure of the Ca²⁺-bound afTMEM16. The methodology is appropriate, and the conclusions appear to be supported by the experimental results. The data support a novel and more reasonable hypothesis about the mechanism of action of afTMEM 16, that can probably be extended to scramblases in general.

Minor point: the biological origin of afTEM 16 could be mentioned in the Introduction.

REVIEWER COMMENTS

We thank the reviewers for their positive evaluation of our work and for their helpful comments and suggestions. Below, we address their concerns.

Reviewer #1 (Remarks to the Author):

While it might seem obvious, the authors should discuss why scrambling does not take place at the dimer interface.

We thank the reviewer for bringing this point up. The idea that lipid scrambling does not occur at the dimer interface was originally proposed by Brunner and Dutzler (Nature, 2014) on the basis that this cavity is hydrophobic. Our results support this notion, as all resolved lipids at the dimer interface and cavity are oriented perpendicular to the plane of the membrane and the membrane appears thicker in this region. This is now discussed on page 9.

The number of experiments should be noted in each legend.

Thanks for pointing this out. We added this information to the legends of Fig. 2, 3, 4 and 5.

Page 4. The authors should also mention that at least two MD simulation papers (Jiang et al., and Bethel et al.) have suggested that membrane thinning contributes to scrambling.

We now mention these two papers on pg. 4.

Page 5. “3 1-palmitoyl-2-oleoyl-sn-glycero-3-phosphoethanolamine (POPE): 1 1-palmitoyl-2-oleoyl-sn-glycero-3-phospho-(1'-racglycerol) POPG nanodiscs 15, α rmsd ~ 0.8 Å.” The sentence is confusing to read. First, better to say 3:1 POPE (...): POPG (...) and make the Ca start a new sentence.

Figure 1A – please label helices.

Figure 1 C,D,E would benefit from some more labels identifying helices.

Thank you for the suggestions! We made all indicated changes.

I am having trouble understanding Fig. 1E because it seems inconsistent with the statement that the extracellular vestibule does not directly interact with lipids.

We thank the reviewer for raising this important point. We added two new figure panels (Fig. 1F-G) where we show a close up of density for the extracellular portion of the pathway and for the lipids. In these panels, we highlight three residues (E305, E310 and R425) that form a salt-bridge are tether the TM3 and TM6 in the extracellular vestibule. We also report in the text (pg. 11) that the phosphate atom of the P3 lipid is 13.8, 17.9 and 15.7 Å away from the side chains of these residues, respectively. We hope these additions clarify our point.

Page 11. I don't understand the sentence “This suggests lipid headgroups only need to traverse the wide intracellular vestibule of the pathway, below the constriction formed by T325 and Y432 (Fig. 2A).” The word traverse implies the conclusion that P3 has moved from the intracellular to the extracellular leaflet, but there is no data to support this.

We apologize for the confusing statement. The point we are trying to convey is that P3 and P4 are the closest resolved lipids with opposite orientation. This suggests that, if scrambling occurs within the groove, then these two lipids respectively define the upper and lower boundary of the portion of the groove that needs to be traversed. We reworded the text to clarify this important point.

Also, the figure does not show Y432

Page 12. “synergistically lower the energy barrier for scrambling” grammar.

Page 13. Abbreviations for POPC and POPG (PC, PG?). Also, there are two ratios here: acyl chain saturation and lipid ratio. So, the phrase “mix of 14:1 lipids” should be clarified.

Figure 4. The groove is open in panel D and closed in panel E. For those who are not cognoscenti of the TMEM16 proteins, the authors should describe these structures more explicitly. Maybe add a cartoon to this figure?

We thank the reviewer for pointing out these mistakes and for the suggestions. We amended Figure 2 to show Y432, Fig. 4 by adding a cartoon of the open and closed groove (new Fig. 4C), and the text of the two confusing paragraphs.

Page 18. The authors do not define MSP1E3 and MSP2N2 nanodiscs in the text – they are only defined in Suppl. Fig. 5.

We apologize for the oversight. MSP1E3 and MSP2N2 are now defined on pg. 5 and 15, respectively.

Page 21. The authors state that there is a discrepancy between the membrane thinning model and previously published MD simulations that show lipids traversing the groove. However, it seems to me that the membrane thinning model does not exclude the possibility that lipids move through the groove - at least some of the time. The groove may not be a specific pathway, but one that has a low energy barrier.

We agree that the membrane thinning model does not exclude that lipids could traverse part of the groove as we explicitly state on pg. 11 and 21. However, our structural data suggests that the extracellular vestibule of the groove does not interact with lipids much, if at all. This is in contrast with MD simulations where this portion of the groove is fully embedded in the membrane and interactions between lipids and E305, E310 and R432 have been proposed to play a key role in scrambling (Bethel and Grabe, PNAS, 2016; Jiang et al., Elife, 2017; Lee et al., Nat Comms, 2018). We hypothesize this difference in protein-lipid interface between MD simulations and cryoEM might play a key role in the discrepancy. This is now more clearly discussed on page 22.

I think the authors should not belabor the equilibration issue.

We toned down this statement. However, we feel it is important to provide at least a plausible origin for the observed discrepancy that could be tested in future computational and wet-lab experiments.

Reviewer #2 (Remarks to the Author):

1. The comparative scrambling rates shown in figure 3 should include at a minimum a statistical t-test between WT and each respective mutant. The statement that the mutations have no effect is somewhat misleading as several of the mutants would appear to have a statistically significant difference at 0 M Ca⁺⁺.

We agree with the reviewer that some mutants show some effects in 0 Ca²⁺. While the effects are statistically significant for all but one mutant (2-tailed t-test, p<0.005), the magnitude of the effects is relatively small (<7-fold in all cases). While our quantitation of the scrambling rate constant captures well large changes, some of the assumptions we make in the analysis make it less well suited to quantify small effects (i.e. <10-fold changes). These limitations are discussed in several of our past publications (Malvezzi et al., PNAS, 2018; Lee et al., Nat Comms, 2018; Falzone and Accardi, MIE, 2020). We now state that the mutants have only minor effects on scrambling. We prefer to refrain from presenting statistical significance of these results as we do not want to convey the idea these effects are more meaningful than they might be.

2. Fig 4A should use nomenclature D511A D514E and not DA/DE
Corrected.

3. Fig 6 is not very informative in the present format. Firstly, the zero height position should be stated in relation to residue within the structure.

We updated the color legend in Fig. 6 to refer the zero-height position of the outer leaflet to H411 and that of the inner leaflet to Q150.

Secondly, this would be more impactful to the reader if this were presented in the plane of the membrane and cut at the single lipid layer and magnified to show the permeation pathway.

Or at least in close up of a single permeation pathway. This reader would prefer both.

We added a supplementary figure (Supp. Fig. 12) where for each nanodisc map we show a cutaway view of the nanodisc density near one permeation pathway.

4. Fig 7 would be enhanced by coloring the two leaflet headgroups different to accentuate the scrambling, i.e. inner leaflet to outer one color and visa versa.

We changed Fig. 7 to distinguish the inner and outer leaflet lipids and to illustrate how scrambling in the various conditions promotes their mixing.

5. The EM data table S1 should list defocus as a positive number by convention. Listing the dose and exposure times to 4 significant figures is both unwarranted and unrealistic that they have been determined to such precision.

6. EM data table S4 again uses unrealistic precision.

7. The EM data table S4 does not list B factors for the protein or lipid. While the lipid densities shown in the supplement are convincing, I believe it is important to list at least the average B factor for the protein and the lipids in this table.

8. There are a small number of typos that should be corrected.

We now use the appropriate convention for defocus values, report one or two significant digits and added the average B factors for the protein, ligands, and waters (where applicable) to the updated version of Table S4.

We thank the reviewer for bringing these omissions and errors to our attention.

Reviewer #3 (Remarks to the Author):

It would be important to provide reviewers with the density maps of the calcium bound afTMEM16

We now share with the reviewers all of our maps used for model building and evaluation of the thickness as well as the deposited pdb models.

1. Please measure the membrane thickness at the dimer interface and also at the lipid pathway, in addition to the distance between the phosphate atoms of the headgroups of P3 and P4, in nanodiscs with C18 lipids, with or without calcium. Given that the lack of support for a conveyor belt mechanism based on their mutagenesis results, it is important to document the membrane thickness around afTMEM16.

While it is tempting to quantify membrane thickness around the protein, we prefer not to do so for several reasons.

- i. The groove does not traverse the membrane perpendicularly, rather it is tilted and curved. Therefore, near the groove there are no points with the same X-Y coordinates with well-defined density for the inner and outer leaflets. This prevents an accurate measurement of thickness in this region. Thus, we prefer to limit ourselves to showing views from the extra-

- and intra-cellular sides (Fig. 6, new Supp. Fig. 12) that illustrate the downward deflection of the membrane near the groove relative to its relaxed plane.
- ii. Measuring thickness near the pathway is affected by arbitrary choices, for example the value of the thickness would change depending on the threshold used to display the nanodisc density as well as on the precise points used for the determination. Further, the maps are at different maximal resolution, so that the strength of density signal close to the protein can vary between them also because of uncertainties on the position of side chains. While normalizing the density by rescaling and resampling the maps together (as we do for the maps in Fig. 6 and in the new Supp. Fig. 12) somewhat minimizes these issues, we do not think these membrane thickness measurements are sufficiently reliable to provide the level of accuracy needed for quantitative comparisons.
 - iii. The groove changes conformation between the open Ca²⁺-bound and closed apo conformations, and the membrane significantly rearranges to accommodate this rearrangement. Therefore, we cannot compare thickness values of the membrane at fixed points relative to the protein.

For these reasons, we believe that maintaining these observations at the qualitative level is more appropriate.

2. The authors found no effects of mutagenesis aimed at disrupting the headgroup interactions of P1-P2, P4-P5-P6 or P2-P5-P6. What about mutagenesis aimed at disrupting the headgroup interactions of P3?

We apologize for the confusion. Mutagenesis for residues coordinating P3 lipid, T325 and Y432, are shown in Fig. 3D-E. We included these in Fig. 3 rather than in Fig. 2 as these residues directly line the open groove and were proposed to form a constriction that had to rearrange to enable lipid permeation (Lee et al., Nat Comms, 2018).

3. Please provide measurements of the membrane thickness around afTMEM16 in nanodiscs with C22 lipids, and D511A/E514A afTMEM16 in nanodiscs with C14 lipids.

Please, see our response to point #1.

4. The nanodisc densities in Fig. 6 appears to show thinning at the rim of nanodiscs. For the main theme of this paper, it is important to quantify the membrane thickness in the vicinity of afTMEM16 beyond P3 and P4.

The thinning at the edge of each nanodisc likely reflects the averaging of many images of finite-size nanodiscs with slightly heterogeneous dimensions. Importantly, the thickness of the nanodisc membrane measured far from the edge is comparable to the values determined with AFM (Table 1), indicating that the contribution of this effect far from the edge is small.

5. Whereas membrane distortion could tilt lipids towards the protein rather than the surface of bilayer, it is important for the authors to demonstrate membrane thinning that facilitates lipid scrambling and make the point clearly in discussion.

We agree with the reviewer on this important point. Indeed, our measurements of scrambling activity in membranes of lipids with different acyl chain length show that membrane thinning does facilitate scrambling. Our structural data shows that (at least in the C18/+Ca²⁺ case) the lipids do rearrange such that their heads tilt towards the protein. These important points are discussed on pg. 9-10 and on pg. 21.

6. On page 14-15, “despite the high resolution of the C22/Ca²⁺ MSP1E3 map, we detect only weak

signals for lipids associated with the pathway-delimiting helices TM4 and TM6.” On page 17-18, “No lipids could be resolved near the closed pathway in these structures. The average resolution of these datasets is lower than that of the two Ca²⁺-bound structures, preventing us from drawing mechanistic inferences from this observation. “ However, the resolution of the structure is 2.3 Å in C18, 2.7 Å in C22 and around 3.1Å for the Ca-free ones. Compare to 2.7 Å, 2.3 Å is significant higher in resolution. It is hard to make such conclusion even between the two Ca²⁺ bound structures. It would help if the authors can share the density maps of the two Ca²⁺ structures, so we can assess how much improvement these maps have made as compared to the published ones with the same condition and how reliable the comparison of lipids is between the two maps.

We agree with the reviewer that it is difficult to draw conclusions on the occupancy of lipids from maps at different resolution, and indeed we only note that in these maps we see weakened/no density for lipids and we state that we cannot draw firm conclusions. We have now toned down our statement to say that the lack of density (pg. 15) “...possibly reflecting a higher energy cost associated with distorting these longer acyl chain lipids”.

We have now shared all maps with the reviewers (and they are all deposited in the EMDB database and the entries will be made public upon publication of the manuscript).

Reviewer #4 (Remarks to the Author):

Minor point: the biological origin of aTfEM 16 could be mentioned in the Introduction.

We thank the reviewer for this suggestion. We now mention that aTfMEM16 is from *Aspergillus fumigatus* on page 4.

REVIEWERS' COMMENTS

Reviewer #1 (Remarks to the Author):

The authors have addressed my concerns satisfactorily.

Reviewer #2 (Remarks to the Author):

The authors have adequately addressed this reviewers concerns and the manuscript is greatly improved. Publication in Nature Communications is recommended.

Reviewer #3 (Remarks to the Author):

The authors have addressed my comments with the revision.